# Assessment of Labour Productivity and the Factors of Its Increase in European Union 27 and Ukrainian Economies

Tetiana Kornieieva *, Miguel Varela, Ana Lúcia Luís and Natália Teixeira

Instituto Superior de Gestão Superior, 1700-284 Lisbon, Portugal
* Correspondence: tetiana.kornieieva@isg.pt or korneevats16@gmail.com; Tel.: +351-591-182-6737

**Abstract:** The article is devoted to identifying the impact of factors on the increasing labour productivity in the economies of the European Union-27 (EU-27) and Ukraine. The system of influencing factors that must be taken into account in the assessment of the labour productivity indicator was substantiated. The factors are based on the most significant indicators of innovative development (innovative activity; formation and the use of personnel; the state of use of fixed capital; the composition of the payroll budget; investment activity; and the use of working time). Based on the use of the method of linearization of the labour productivity model, the rating coefficients of the influence of factors for the economies of the EU-27 and Ukraine were determined. It has been proven that the following factors have a significant positive impact on labour productivity: an increase in the costs of scientific research and development; the growth of enterprises expenditures on research and development (R&D) in high-tech sectors; the increase in the share of scientific research personnel and researchers in the total number of the employed population; the growth of costs for the innovation of industrial enterprises; an introduction to the production of new technological processes and innovative types of products; the purchase of machinery, equipment, and software; an increase in the share of the employed population that has a basic higher education (bachelor's degree); the growth in the share of the employed population that has a full higher education (master's degree); the increase in the share of enterprises providing training; the increase in the capital–labour ratio; the technological equipment of labour; the machine equipment of labour; the renewal of fixed capital; and the increase in the level of intellectualization of fixed capital. The available reserves for increasing the labour productivity in EU-27 economies and Ukraine were clarified, and recommendations for managing the labour productivity in the conditions of innovative development have been provided. This study gains relevance when Ukraine has assumed the official status of an EU candidate country and the advantages and potential difficulties in the membership process should be evaluated. Labour productivity will be one of the key factors in the post-conflict recovery of the economy.

**Keywords:** factors; assessment; labour productivity; innovations; personnel; fixed capital; payroll budget; investment; reserves for increasing labour productivity

**JEL Classification:** B22; E24; O47

## 1. Introduction

The problem of ensuring the growth of labour productivity become especially urgent in connection with the need to increase the competitiveness of national economies. Labour productivity determines the level and quality of social life, as an increasing labour productivity affects the level of peoples' salary, social benefits, pensions, the ratio of prices and costs, and inflation control, according to Yi and Chan (2014). It also determines the competitiveness of products at the international and national levels, provides the possibility of structural reforms of the economy on the newest innovative and technological basis (Bhawsar and Chattopadhyay (2015), Cantwell (2006), Tether and Hipp (2002), Fagerberg (1996), and Buckley et al. (1988)). The higher the level of labour productivity, the stronger

the economic potential of a country is. The richer the society, the more opportunities for the growth of the peoples' well-being there is. The stability of the economy, overcoming the consequences of a crisis, and the transition to economic growth depend on the level of labour productivity.

A comparative analysis of the level of labour productivity between EU-27 economies (from 2020) and Ukraine is the basis for making decisions on investment, innovation and technical reforms of production, the management of labour resources, the determination of price policy, the assessment of profitability and competitiveness, and the regulation of foreign economic relations.

There are factors influencing the increase in labour productivity which need to be studied in order to carry out a comprehensive analysis of the social and economic development of national and international economies. The assessment of labour productivity and the identification of reserves for its growth on the example of a comparative analysis of the economies of the EU-27 and the economy of Ukraine remains one of the key problems.

**Analysis of recent research and publications.** The problem of labour productivity has become the subject of study for many foreign researchers: Gilbert (2013), Marx and Engels (1975), Ricardo (2001), Bastiat (2010), Smith (2018), Drucker (2003), AlKathiri (2022), Verdoorn (2002), and Gomez-Salvador et al. (2006), amongst others.

Marx and Engels (1975) understood "productive labour" as any hired labour that creates added value for a capitalist. According to Smith (2018), the national annual product can be increased only by increasing the number of productive workers and increasing their productivity. Ricardo (2001), as one of the most famous representatives of the school of classical political economy, interpreted labour productivity as the most important factor influencing income. Ricardo (2001) believed that the economic goal of society was to increase the profits of capitalists, and, therefore, to increase labour productivity. The evolution of views on the category of "labour productivity" was manifested due to the emergence of the theory of "factors of production", formulated by the French scientists Say (2001) and Bastiat (2010). The current problems of labour and capital productivity were considered by analysing productivity as a necessary condition for international competition. Nader AlKathiri (2022) used the method that allows for determining the growth of labour productivity by technical and technological components. Gomez-Salvador et al. (2006) studied the factors affecting the development of labour productivity in the EUR zone to ensure support for economic growth.

In Ukraine, problems of labour productivity management were fruitfully studied by Grishnova et al. (2015), Kolot and Herasimenko (2020), Libanova (2019), Fomishin and Mochernyi (2010), Plaksov (1998), Gavkalova and Zolenko (2019), Semykina et al. (2021), Cherep et al. (2019), and Shaulska et al. (2021).

Ukrainian researchers, in particular Fomishin and Mochernyi (2010), interpreted labour productivity as "the efficiency of people's production activities in the process of creating material goods and services". From the point of view of Grishnova et al. (2015), labour productivity is "a general indicator of the use of labour force, which, like all efficiency indicators, characterizes the ratio of results and costs, in this case—results and labour costs". According to Semiv (2013), labour productivity is "an important indicator that characterizes the ability of the economic system to function with proper efficiency".

The factors affecting labour productivity are classified differently by researchers. Lepeyko (2015) provided a classification that involves the division of the labour productivity growth factors into six groups: a change in the nomenclature and the range of products; an increasingly technical level of production; the organizational factors; a change in the production volume; and the natural factors. These factors are relevant mainly for agriculture and tourism. Paseka (2010) divided the factors which affect the level of labour productivity into researcher groups: technical–organizational, social–economic, and natural–climatic. Some researchers have distinguished only three groups of factors: the natural conditions; the subjective factors; and the economic factors (Fomishin and Mochernyi 2010).

Several studies have been conducted concerning labour productivity in the EU. Marelli and Signorelli (2010) identified the main growth models for EU countries, focusing on the employment–productivity relationship and the main determinants of productivity. The conclusion is that high employment growth is likely to lead to a slower productivity growth, with the main determinants of productivity being education, a transition index, some structural indicators, and what the authors called a "shadow economy" proxy (Marelli and Signorelli 2010).

The study of Kutan and Yigit (2009) estimated the determinants of labour productivity growth with an emphasis on the impact of globalization and the EUs integration efforts on labour productivity growth. Concerning the internal variables, human capital is considered to be the most relevant to justify the labour productivity growth in the newer member states, and there is also a considerable "catching up" effect, and hence a real convergence.

Pariboni and Tridico (2020) explained the reasons behind the dynamics of labour productivity growth during a process of change: the speed of investment, which incorporates innovation; the speed of research and development, which enables the emergence of new ideas, showing the dynamism of society; the deregulation of labour markets and the use augmentation of temporary employment; and the direction set by structural change itself.

Inklaar et al. (2005) found that aggregate labour productivity growth can be divided into contributions from labour quality, ICT, non-ICT capital deepening, and total factor productivity. Timmer and Van Ark (2005) compare the effects of information and communication technologies on aggregate labour productivity growth, particularly as a factor in the total factor productivity growth, which partially justifies the USAs lead in labour productivity growth.

Education and professional training make the work of each individual more productive (Blundell et al. 1999). Education either increases a worker's productivity at the workplace or makes him/her capable of such work, the result of which is receiving a better pay. Accordingly, an increase in the qualification and level of education of an economically active population increases labour productivity in the economy. On the other hand, education contributes to the development of personal business skills and entrepreneurship (Gorman et al. 1997); (Gibb 1993). According to Lange and Topel (2006), education significantly increases social labour productivity, stimulates economic growth, and shortens the time of the dissemination of scientific and technical discoveries. Education is the determining factor in the political, socio-economic, cultural, and scientific activities of a society and state (Astakhova et al. 2016).

Egger and Egger (2006) analysed the role of international outsourcing on the productivity of low-skilled workers in the EU production process, concluding that in the short run, data show a negative marginal effect on the real value which is added per worker, while in the long run, it shows a positive impact. "This may be explained by imperfections in European labour and goods markets, which prohibit an immediate adjustment in the factor employment and the output structure" (Egger and Egger 2006, p. 98).

Klein and Ventura (2009) analyse the main barriers to labour mobility across countries that tend to coexist with relevant differences in living standards, partially due to productivity differences. The model based on endogenous labour movements is used as a tool to assess the effects of removing barriers to labour mobility on production, capital accumulation, and an increased societal welfare (Klein and Ventura 2009). Considering the relationship of labour with capital, Autor and Salomons (2018, p. 1) analysed if the capital–labour substitution need, which is normal in developed countries, does not reduce the aggregate labour demand because it can provoke "own-industry output effects; cross-industry input-output effects; between-industry shifts; and final demand effects".

The analysis of scientific research shows that a single point of view has not yet been formed regarding the definition of factors that have a greater influence on labour productivity at the macroeconomic level. Mainly, there is no universally recognized methodological approach to the identification of labour productivity growth reserves at different economic levels.

**Previously unsolved parts of the overall problem.** Issues related to systematic studies of the main factors and the search for reserves for the formation of new conceptual approaches and methods of increasing labour productivity and ensuring the effective management of the socio-economic processes at various economic levels of EU-27 economies and Ukraine's economy remain insufficiently studied.

**The objective of the research** is to develop scientific and practical recommendations for increasing labour productivity in the economies of EU-27 counties and Ukraine.

**Main material.** Labour productivity is the main criterion for the efficiency of an economy, simultaneously forming the economic basis for raising the standard of living of the population. The increase in labour productivity is extremely important for the achievement of the socio-economic standards of life in society which are recognized by world community because it is labour productivity that is the fundamental basis of economic growth.

The importance of increasing labour productivity was substantiated in the UN Millennium Declaration, approved by 189 countries in 2000 at the UN Millennium Summit. It was then that the goal was declared: by 2015, the world community is to achieve the results in those areas where the unevenness of global human development turned out to be most acute. However, the problem of defining clearer guidelines for world development after the achievement of the Millennium Development Goals in 2015 became gradually relevant. Adopted by the UN General Assembly, the 17 Sustainable Development Goals (SDGs) for the period up to 2030 have given a new impetus to the global efforts to achieve a sustainable development. The EU plays an active role in helping to maximise progress towards the SDGs. Encouraging sustainable economic growth by achieving higher levels of productivity through technological innovation is one of the most important goals of sustainable development.

The 2030 Agenda for Sustainable Development and its 17 SDGs, adopted by the UN General Assembly in September 2015, have given a new impetus to global efforts for achieving a sustainable development. The EU is fully committed to playing an active role in helping to maximise progress towards the SDGs (monitoring report on progress towards the SDGs in an EU context, Sustainable Development in the European Union 2022).

Labour productivity, as an economic and social category, is important when analysing the development of countries, which reflects the standard of living of the population, the rates of economic growth, and the state of competitiveness of the national economy. At the same time, one of the priorities for the development of the economy on an innovative basis should be an increase in labour productivity under the conditions of an active implementation of modern scientific and technical achievements in the production and intensive formation of the sector of highly intelligent services, which is capable of ensuring a high level of added value, population income, investments, stimulating human development, and the process of forming high social standards.

Increasing labour productivity is one of the priority directions for the development of the economy of industrialized countries, oriented, first of all, around the use of skilled labour, which is based on the latest knowledge, the achievements of science and technology, the use of mechanization, automation, computerization, etc.

Productivity is one of the indicators of the competitiveness of the EU economy and the ability to ensure the well-being of its citizens. Technological and organizational innovations, as well as improving the skills of workers, are among the main factors that ensure a productivity growth. If the gross domestic product (GDP) increases and the number of hours worked remains the same, then productivity increases. This is expressed in the growth of the volume of production per hour of work. The GDP in constant prices is calculated by the purchasing power parity (PPP) in relation to the average value for the European Union, which allows for a correlating productivity in different EU countries. Data are collected using high methodological standards. The limitations in comparability are due to the fact that the total number of working hours differs in different European countries.

Productivity issues are addressed in the First Lisbon Strategy. Initially, the Lisbon Strategy was based on the resolution of the European Council (2000), which aimed to

transform the EU into the most competitive knowledge-based economy in the world by 2010. A key role in formulating the strategy was played by the Portuguese economist Maria João Rodríguez. The implementation of the strategy was aimed at ensuring a strong and long-term economic growth and creating more jobs, which should ultimately make Europe a comfortable place for investment and work. Additionally, the Lisbon Strategy was developed to solve such problems as the aging population, the increase in productivity, and the increase in competitiveness in the context of economic globalization. The initial action plan was agreed upon by all EU member states. It included increasing investments in education, research, and innovation in order to increase the welfare and level of protection of its citizens.

An increase in funding for the development of the market, modern infrastructure, and the reduction in bureaucracy was foreseen in order to create favourable conditions for companies that introduce innovative technologies and create jobs, as well as to ensure environmental safety (Varela and Firmino 2015).

The results of the implementation of the Lisbon program in 2005 were disappointing, especially in the field of employment. Therefore, the European Council decided to primarily focus its attention on the creation of new jobs. In order to give this program a powerful new impetus, the European Commission established a simplified coordination procedure and focused on National Action Plans (NAPs). The emphasis was shifted towards the adoption of more operational measures which were needed by the EU member states.

The official signing of the Lisbon Agreement took place on 13 December 2007. The agreement was approved by all 27 EU countries, and it entered into force on 1 December 2009.

In the period 2000–2010, the Lisbon Strategy became the most successful action plan for the development of the European Union, despite the fact that its long-term goals have not been fully achieved today.

At that time, there was an understanding that this strategy should also be adopted for the period of 2010–2020. The need to modernize the economy became obvious after the economic crisis, which pointed to numerous structural deficiencies in the economies of the EU member states. The new "Europe 2020" strategy began to be implemented in March 2010. Special attention was paid to the problems of innovation and the increasing growth rates in relation to the issues of employment and social cohesion.

Several research centres on labour productivity exist in Ukraine: the Ukrainian Research Institute of Productivity of Agro-Industrial Complex, zonal and regional centres of productivity, which are subordinate to the Ministry of Agrarian Policy and Food of Ukraine; the Research Institute of Social Policy of the Ministry of Social Policy of Ukraine and the National Academy of Science of Ukraine; and the Institute of Industrial Economics of the National Academy of Sciences of Ukraine.

At the macro level, labour productivity growth manifests itself through an increase in GDP volumes, which forms the economic basis for improving the quality of life of the population, solving social problems, ensuring an expanded production, and securing the highly competitive position of countries in the world markets.

The assessment of labour productivity in EU-27 economies (as of 2020) indicates an increase in this indicator for the period of 2014–2021 by 11.6% from 62,014.1 EUR/person (2014) to 69,186.1 EUR/person (2021), (Figure 1). This is explained by the fact that the rate of the decrease in the number of the employed population in the EU-27 economy exceeds the rate of GDP reduction. Thus, there was a reduction in the number of the employed population in the EU-27 economy by 13.7% from 241.9 million people (2019) to 208.8 million people (2021) and a decrease in GDP by 12.4% from EUR 16,491,885.3 million (2019) to EUR 14,447,940.6 million (2021). This situation is connected with the growth of the share of the population who are of retirement age and the reduction in the share of those who are youths and persons of working age. Additionally, the decrease in the number of the employed population and GDP is associated with the exit of the United Kingdom from the European Union in 2020. The following factors significantly influence the increase in labour

productivity in EU-27 countries: the introduction of new technologies; the increasing professional level of managers; systematic professional development, professional education; investment in personnel development; and others.

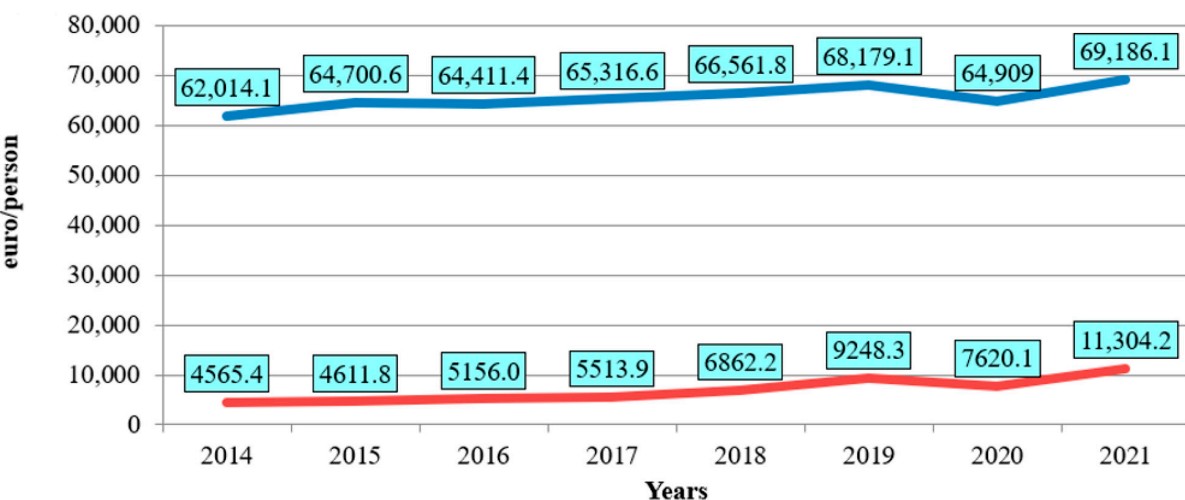

**Figure 1.** Indicators of labour productivity at the level of the economies of the European Union-27 countries and Ukraine, 2014–2021. Source: calculated and plotted by the authors according to data. (European Commission Eurostat 2022; Organisation for Economic Co-Operation and Development 2022; World Bank 2022).

Regarding the Ukrainian economy, labour productivity has increased by an astonishing 147.6%, from 4565.4 EUR/person (2014) up to 11,304.2 EUR/person (2021), (Figure 1). This was due to an increase in the level of GDP at the purchasing power parity (PPP) of Ukraine by 27.4% and a decrease in the number of employed people in the economy by 13.2%, from 18.073 million people (2014) to 15.693 million people (2021).

The comparison is straightforward. Despite the surprising increase in labour productivity in Ukraine, the figures are still far from desirable. The increases seen in the EU-27 are much smaller, as expected in more mature economies. Ukraine will have to be able to sustain this rate of growth in the post-conflict period if it is to come close to the EU-27 average.

## 2. Model

Nowadays, the issue of determining the influence of a number of socio-economic, investment, innovation, technical and technological, and organizational factors on the level of labour productivity is very significant.

In order to solve it, it is necessary to develop a methodology for building a model for calculating the rating coefficients of the influence of economic development factors at the macro-level on labour productivity.

The modern methods of assessing labour productivity, including taking into account the resource potential, were studied. It was found that they determine only the impact of certain factors on the efficiency of using certain resources: the correlation relationship, the multiple linear regression model, the correlation-regression analysis method, the Ferrar–Glouber algorithm, the Ridge estimation method (i.e., Ridge regression), the method of extrapolation, and the method of systematization. However, a certain part of the theoretical and practical issues related to the determination of the influence of the factors

of economic development at the macro-level on labour productivity have not yet been fully considered. In particular, the issue of determining the rating of the influence of the economic development factors at the macro level on labour productivity has not been sufficiently studied.

To develop the methodology, the most influential (hypothetically) factors of economic development at the macro level, which are listed in Tables A2 and A3, were selected.

Each of the factors listed in the table has a different effect on labour productivity. The level of influence can be defined in a numerical expression.

This methodology can be used at the macro, meso, and micro levels (Korneeva 2020).

In the model, each indicator within a separate group of influencing factors got the notation $X_i$ (within the model $X_1$–$X_{27}$, since exactly 27 indicators were studied and identified, within 5 groups of factors). They are listed in Tables A2 and A3.

Each of the factors listed in the table has a different effect on labour productivity in EU-27 and Ukraine economies. The level of influence can be defined in a numerical expression.

To determine the rating indicator of the influence of factors on labour productivity, we introduce the concept of the factor rating coefficient, which can be the coefficient for the factor $X_i(B_i = tg\varphi_i)$ of a linear model. That is, the larger the angle $\varphi_i$ between the linearized functional and the abscissa, the faster the labour productivity indicators increase when the factor value changes. In fact, $B_i$ is a criterion of the significance of the factor $X_i$ (Figure 2).

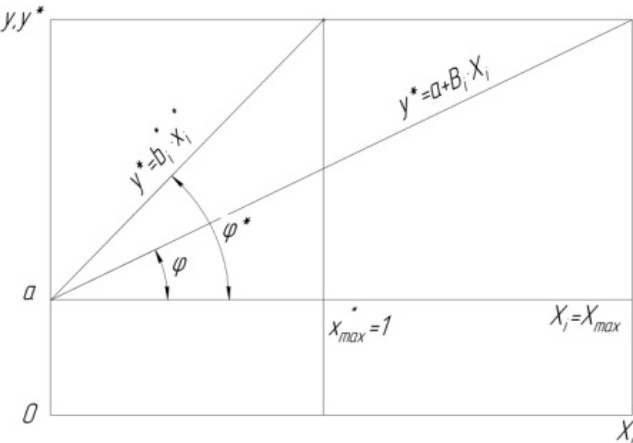

**Figure 2.** Transferring natural values of factors ($X_i$) to code (dimensionless) forms ($x_i$) and transferring the beginning of the ordinate to a point: $y = a$: $y = a + B_i \times X_i$ —is a functional (labour productivity at the level of the economies) for natural values of factors; $y = a + b_i \times x_i$ —is a functional for the coded values of factors.

In the linear polynomial (functional), according to which the labour productivity indicator is determined:

$$y = B_0 + B_1 \times X_1 + B_2 \times X_2 + \ldots + B_i \times X_i + \ldots + B_k \times X_k + B_{12} \times X_1 \times X_2 + B_{13} \times X_1 \times X_3 + \ldots + B_{ij} \times X_i \times X_j + \ldots + B_{n-1,n} \times X_{n-1} \times X_n \tag{1}$$
$$i \neq j$$

the influence of the paired interactions $(X_i \times X_j)$ is not taken into account due to the impossibility of managing the changes in the factors of economic development in given intervals in real production conditions.

In the generally accepted format of dependence $y_i = f(X_i)$, it is not possible to compare the coefficients $B_i(tg\varphi_i)$ with each other, since the scales of the values of the factors on the axis $X_i$ are different and depend on the units of measurement.

To enable the comparison of the criteria of the significance of the factors, they must be presented in a coded (dimensionless) form.

For a two-dimensional linear model:

$$y = a + B_i \times X_i \tag{2}$$

where $X_i$ is the abscissa (the factor in the natural expression with the corresponding unit of measurement). After moving the beginning of the ordinate to the point $y = a$ (Figure 2) we shall consider the values of the factors in the code form $x_i^*$ with the interval of the values for all factors $(x_i^* \dots x_k^*)$ from «0» to «1».

After the transformations of formula (2) in the new coordinate system, we shall get the value $b_i^*$ (rating coefficient of the $i$-th factor in the comparison format) in a single scale $x_i^*(0 - 1,$ equivalent to the natural value $X_i)$. For the calculations, we shall accept the correspondence $X_{i\,max} \to x_{i\,max}^*$, where, according to the condition $\left(x_i^* = 0 \dots 1\right)$ $x_{i\,max}^* = 1$.

$$\text{Then } b_i^* = B_i \times X_{i\,max}. \tag{3}$$

The value of the rating coefficient does not indicate the absolute level of its influence. It is used to establish the relative impact of the factors on functionality, labour productivity, ($y$) and their ranking.

For the practical verification of the developed methodology and analysis of the research results, as an example, we have chosen the main components of the development of the countries' economies: the investment activity; the innovative activity; the state of use of fixed capital; the formation and use of personnel; the composition of the payroll budget; and the use of working time.

The values of the rating coefficients of the influence of the significant factors of economic development for the EU-27 economies on labour productivity ($y$), calculated according to the presented values of the factors in the code (dimensionless) form, are given in Table 1.

To determine the level of influence of the groups of economic development factors for the EU-27 economies on labour productivity, the average values of the rating coefficients, which are given in Table 1 (column 5), are calculated according to the data in Table 2.

To determine the level of influence of the groups of the economic development factors for the Ukrainian economy on labour productivity, the average values of the rating coefficients, which are given in Table 3 (column 5), are calculated according to the data in Table 4.

**Table 1.** The value of the coefficients of the rating of the influence of economic development factors in the EU-27 economies on labour productivity (*y*).

| Factor | Name of the Factor | Equation $y_{ji} = B_i \times X_i + a_i$ | The Maximum Value of the Abscissa $X_{imax}$ | Rating Coefficients (RC) of Factors $K_i = b_i^* = B_i \times X_{imax}$ |
|---|---|---|---|---|
| 1 | 2 | 3 | 4 | 5 |
| $X_{18r\&d}$ | Research and development expenditures | 3.548x + 1264.1 | 92.818 | 329.318 |
| $X_{31}$ | Share of employed population with a complete higher education (master's degree) | 9.8636x − 51.598 | 28.827 | 284.338 |
| $X_{18ber\&d}$ | Business enterprises R&D expenditure in high-tech sectors | 103.75x − 29.071 | 2.677 | 277.739 |
| $X_{30}$ | Share of employed population with basic higher education (bachelor's degree) | 5.1818x − 55.476 | 52.014 | 269.526 |
| $X_{18\_r\&dehtse}$ | R&D expenditures in high-tech sectors per enterprise | 3.1119x − 13.732 | 83.169 | 258.814 |
| $X_{18sr\&d}$ | Share of R&D expenditures in GDP | 3.0473x − 125.29 | 78.400 | 238.908 |
| $X_{18ems}$ | Enterprises in high-tech manufacturing sectors | 0.5905x − 18.333 | 394.000 | 232.657 |
| $X_{18\_sr\&dps}$ | Share of R&D personnel and researchers in total employed population | 0.802x − 12.502 | 286.400 | 229.693 |
| $X_{31\_pret}$ | Participation rate in education and training by employed population | 0.9538x − 14.728 | 239.600 | 228.339 |
| $X_{31\_ept}$ | Share of enterprises providing training | 5.494x + 72.485 | 40.123 | 220.424 |
| $X_{1sap}$ | The share of the active part of fixed capital | 2.4163x + 16.104 | 85.000 | 205.360 |
| $X_2$ | The share of machinery and equipment in the active part of fixed capital | 0.0025x + 21.302 | 76.917 | 192.293 |
| $X_8$ | Intellectualization coefficient of fixed capital | 2.074x + 23.339 | 90.000 | 186.660 |
| $X_3$ | Capital–labour ratio | 19.8037x + 234.73 | 9.146 | 181.125 |
| $X_4$ | Technological equipment of labour | 1.9935x − 47.845 | 89.474 | 178.366 |
| $X_5$ | Machine equipment of labour | 1.999x + 39.797 | 89.000 | 177.911 |
| $X_9$ | Capital investment. total | 1.6718x − 20.864 | 100.000 | 167.180 |
| $X_{9si}$ | Share of investment in GDP | 1.5233x − 25.792 | 99.900 | 152.178 |
| $X_{22\_ptelpe}$ | The coefficient of part-time employment due to the lack of a principal place of employment | 3.4397x − 57.988 | 43.972 | 151.252 |
| $X_{22\_pter}$ | The coefficient of part-time employment in relation to the total employment | 2.3903x + 131.71 | 62.052 | 148.322 |
| $X_{22\_ewtse}$ | The coefficient employment of working time of skilled employees to all employees in the industry | 2.6007x + 21.724 | 55.682 | 144.812 |
| $X_{22\_ptet}$ | The coefficient of part-time employment due to education, training | 2.5268x + 49.128 | 35.385 | 89.411 |

Source: calculated and compiled by the authors based on the European Commission Eurostat (2022), Organisation for Economic Co-Operation and Development (2022), World Bank (2022).

**Table 2.** The average values of the rating coefficients of the influence of economic development factors in the EU-27 economies $K_i$ (by the groups) on labour productivity ($y$).

| № | Groups of Factors | Notation of Factor Group Rating Coefficients | Number of Factors (Plan) $N_i$ | Number of Influential Factors ($N_{inf}$) | Number of Positively Influencing Factors ($N_{pi}$) | $\sum K_i$ Total Rating Factor | | $\overline{K_u} = \frac{\sum_{i=1}^{N_u} K_i}{N_u}$ Average Values of Rating Coefficients | |
|---|---|---|---|---|---|---|---|---|---|
| | | | | | | of All Factors | of Positively Influencing Factors | of All Factors | of Positively Influencing Factors |
| 1 | 2 | 3 | 4 | 5 | 6 | 7 | 8 | 9 | 10 |
| 1 | Innovation activity | $RC_{innov}$ | 6 | 6 | 5 | 1995.177 | 1757.078 | 249.397 | 251.011 |
| 2 | Formation and use of personnel | $RC_{person}$ | 4 | 4 | 3 | 1395.957 | 1395.957 | 199.422 | 215.582 |
| 3 | State of use of fixed capital (FC) | $RC_{fc}$ | 6 | 6 | 5 | 569.931 | 569.931 | 189.977 | 201.140 |
| 4 | Investment activity | $RC_{invest}$ | 2 | 2 | 2 | 515.886 | 511.105 | 103.177 | 103.177 |
| 5 | Use of working time | $RC_{wt}$ | 4 | 4 | 3 | 560.519 | 560.519 | 80.074 | 93.420 |
| 6 | Coefficient for all groups of factors | $C$ | 22 | 22 | 18 | 5037.470 | 4794.590 | 140.284 | 177.577 |

Notes. 1. $y_1$—is labour productivity at the level of the economies of the European Union-27 countries, EUR/person; $K_i$—$i$-th rating coefficient. 2. $\sum K_i \overline{K_u} = \frac{\sum K_i}{N}$—are the sum of the rating coefficients and the average rating coefficient (by factor groups), respectively. 3. $N_i$¯is the total number of factors (by groups), $N_{inf}$¯is number of influential factors, $N_{pi}$¯is the number of positively influential factors. 4. Calculated and compiled by the authors according Table 1.

**Table 3.** The value of the coefficients of the rating of the influence of economic development factors in the Ukrainian economy on labour productivity ($y$).

| Factor | Name of the Factor | Equation $y_{ji} = B_i \times X_i + a_i$ | The Maximum Value of the Abscissa $X_{imax}$ | Rating Coefficients (RC) of Factors $K_i = b_i^* = B_i \times X_{imax}$ |
|---|---|---|---|---|
| 1 | 2 | 3 | 4 | 5 |
| $X_1$ | Fixed assets in the economy at actual prices | 4.3471x − 26.456 | 70.464 | 306.314 |
| $X_{30}$ | The share of employed population with basic higher education (junior bachelor's, bachelor's degree) | 5.5531x − 58.365 | 50.844 | 282.340 |
| $X_3$ | Capital–labour ratio | 13.76x − 63.094 | 20.0 | 275.200 |
| $X_6$ | Renewal coefficient | 3.5848x − 28.711 | 73.0 | 261.690 |
| $X_{31}$ | The share of employed population with a complete higher education (master's degree or qualification level of a specialist) | 3.4066x − 17.17 | 73.0 | 248.682 |
| $X_{32}$ | The share of employed population with incomplete higher education (qualification level of a junior specialist) | 0.336x − 14.037 | 727.0 | 244.506 |
| $X_{18eie}$ | Expenditures on innovation of industrial enterprises | 0.4706x − 0.7697 | 510.0 | 240.006 |
| $X_{18r\&d}$ | R&D expenditure—total | 0.5583x − 0.8381 | 427.2 | 238.506 |
| $X_{18r\&die}$ | Expenditures on R&D of industrial enterprises | 227.946x + 10.702 | 1.031 | 235.012 |
| $X_{18nt}$ | The acquisition of new technologies (acquisition of other external knowledge) | 6.0894x − 17.387 | 37.859 | 230.539 |
| $X_{17}$ | The number of new technological processes introduced into production | 2.8674x − 69.706 | 78.316 | 224.567 |
| $X_{18}$ | The number of introduced innovative types of products, by names | 8.248x + 31.025 | 26.595 | 219.367 |
| $X_{18mes}$ | The acquisition of machinery, equipment and software | 0.0019x − 3.0596 | 114617.0 | 217.772 |
| $X_{18r\&dp}$ | Share of R&D expenditures in GDP | 3.1714x − 1.4665 | 59.638 | 189.136 |
| $X_{25}$ | The share of basic salary in the payroll budget | 1.6353x − 54.708 | 100.0 | 163.530 |
| $X_{26}$ | The share of additional salary in the payroll budget | 3.3801x − 130.58 | 47.680 | 161.167 |
| $X_{28}$ | The share of payment for the time not worked in the payroll budget | 13.5406x + 58.901 | 9.598 | 129.963 |
| $X_{26}$ | The share of additional salary in the payroll budget | 4.0017x + 186.41 | 30.900 | 123.651 |
| $X_{10}$ | The share of investments in fixed capital to the total amount | 1.3951x − 22.808 | 87.4 | 121.932 |
| $X_9$ | Capital investment. total | 1.193x + 31.333 | 100.0 | 119.297 |
| $X_{11}$ | The share of investments in capital construction to the total amount | 1.6848x + 103.7 | 54.211 | 91.335 |
| $X_{12}$ | The share of investments in machinery, equipment and inventory to the total amount | 6.723x + 90.694 | 13.256 | 89.126 |
| $X_{14}$ | The share of investments in capital repairs to the total amount | 0.9276x + 631.19 | 81.547 | 75.646 |
| $X_{27}$ | The share of incentive and compensation payments in the payroll budget | 32.995x + 180.79 | 2.039 | 67.277 |
| $X_{15}$ | The coefficient of intellectualization of fixed capital investment | 0.638x + 75.502 | 92.903 | 59.281 |
| $X_{29}$ | The average monthly nominal salary per employee | 0.5404x + 68.3 | 13.323 | 7.201 |
| $X_7$ | Coefficient of wear | 0.2328x + 210.46 | 5.287 | 1.231 |

Source: calculated and compiled by the authors based on the European Commission Eurostat (2022), Organisation for Economic Co-Operation and Development (2022), World Bank (2022), State Statistics Service of Ukraine (2022).

**Table 4.** The average values of the rating coefficients of the influence of economic development factors in the Ukrainian economy $K_i$ (by the groups) on labour productivity ($y$).

| № | Groups of Factors | Notation of Factor Group Rating Coefficients | Number of Factors (Plan) $N_i$ | Number of Influential Factors ($N_{inf}$) | Number of Positively Influencing Factors ($N_{pi}$) | $\sum K_i$ Total Rating Factor | | $\overline{K_u} = \frac{\sum_{i=1}^{N_u} K_i}{N_u}$ Average Values of Rating Coefficients | |
|---|---|---|---|---|---|---|---|---|---|
| | | | | | | of All Factors | of Positively Influencing Factors | of All Factors | of Positively Influencing Factors |
| 1 | 2 | 3 | 4 | 5 | 6 | 7 | 8 | 9 | 10 |
| 1 | State of use of fixed capital (FC) | $RC_{fc}$ | 4 | 4 | 3 | 785.572 | 397.201 | 261.857 | 289.490 |
| 2 | Formation and use of personnel | $RC_{person}$ | 3 | 3 | 3 | 1641.222 | 1641.222 | 234.460 | 234.460 |
| 3 | Innovation activity | $RC_{innov}$ | 9 | 9 | 9 | 556.987 | 556.987 | 139.247 | 139.247 |
| 4 | Composition of the payroll budget | $RC_{pb}$ | 5 | 5 | 5 | 807.618 | 807.618 | 115.374 | 115.374 |
| 5 | Investment activity | $RC_{invest}$ | 6 | 6 | 6 | 11.228 | 11.228 | 10.071 | 10.071 |
| 6 | Coefficient for all groups of factors | $C$ | 27 | 27 | 26 | 3802.627 | 3414.256 | 183.627 | 219.452 |

Notes. 1. $y_1$—is labour productivity at the level of the national economy of Ukraine, EUR/person; $K_i$—$i$-th rating coefficient. 2. $\sum K_i$ и $\overline{K_u} = \frac{\sum K_i}{N}$—are the sum of the rating coefficients and the average rating coefficient (by factor groups), respectively. 3. $N_i$—is the total number of factors (by groups), $N_{inf}$—is number of influential factors, $N_{pi}$—is the number of positively influential factors. 4. Calculated and compiled by the authors according to Table 3.

### 3. Results and Discussion

According to the developed model at the macroeconomic level, we calculated the rating coefficients of the influence of the significant factors on labour productivity ($y$) for the main components of the country's economic development: investment activity; innovative activity; the state of the use of fixed capital; the formation and use of personnel; the composition of the payroll budget; and the use of working time.

One of the most influential groups of factors on labour productivity in EU-27 countries is "Innovative activity", which is characterized by an increase in:

- Research and development expenditures ($X_{18\_r\&d}$) by 22.6% from 2014 (EUR 286,510.479 million) to 2020 (EUR 351,364.496 million);
- The amount of expenditures of enterprises for research and development (R&D) in high-tech sectors ($X_{18\_ber\&d}$) by 27.3% from 2014 (EUR 182,357.415 million) to 2020 (EUR 232,094.171 million);
- The amount of expenditure on R&D in high-tech sectors per enterprise ($X_{18\_r\&dehtse}$) by 23.5%. For example, in 2014, R&D spending per enterprise in high-tech sectors was EUR 86.0 million, and in 2020 it was EUR 106,230.0 million (Figure 3).

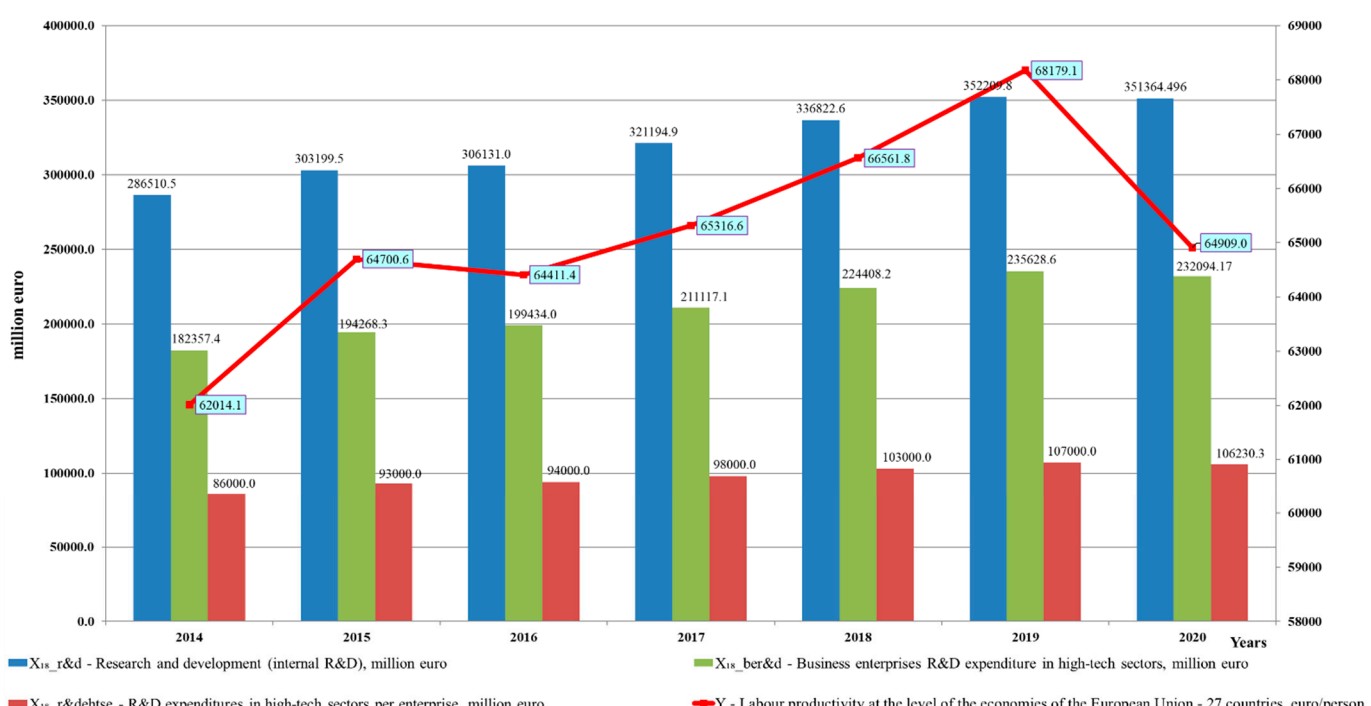

**Figure 3.** Innovation activity in the EU-27 economies, in 2014–2020. Source: calculated and plotted by the authors according to data (European Commission Eurostat 2022; Organisation for Economic Co-Operation and Development 2022; World Bank 2022).

A positive phenomenon is the growth of the number of enterprises in high-tech production sectors in European Union-27 countries, in 2014–2020, by 3.1%.

Over the past 7 years, the share of research staff and researchers in the total number of the employed population ($X_{18\_sr\&dpr}$) in EU-27 countries has increased by 0.24 percentage points (p.p.), from 1.319% (2014) to 1.559% (2020), which affected the growth of the labour productivity, reaching out maximum values in 2018 of 66,561.8 EUR/person, in 2019 of 68,179.1 EUR/person, and in 2021 of 69,186.1 EUR/person. Highly qualified specialists are the most important subjects in the field of innovation. Skilled human resources, combined with the environment that encourages intensive R&D learning processes, can combine prior knowledge and gain new opportunities, as well as stimulate innovation and creativity. The dynamics of education and the training of human resources is a key element of

innovation and determines the innovative capacity of territories. One of the determinants of the innovative capacity is the creation of new knowledge, which can be stimulated by increasing government and administrative expenditures on R&D, as well as through investments in information and communication technologies. This affects the innovative products, stimulating their differentiation, stimulating the development of new market niches, and enabling the introduction of new technological products. Countries interact with each other to influence productivity, economy, politics, and culture in their close relationship with the development of information technology. Financial resources are both limitations and drivers of innovation development, therefore they play an important role in forming the innovation potential.

The activation of innovative potential and the implementation of scientific and research works can be realized due to the combination of government subsidies with private investments.

A new vision for the European Research Area (ERA) aims to build a common scientific development and technology for the EU, by priorities in investments and reforms, improving access to excellence, directing research, and innovation results into the economy and deepening the ERA. The EU has a long-standing objective of increasing its R&D intensity to 3% of GDP, which was reaffirmed in a Council Recommendation on the Pact for Research and Innovation in Europe from November 2021.

The EU research and innovation programme Horizon Europe aims to support researchers and innovators to drive the systemic changes needed to ensure a green, healthy, and resilient Europe.

The EU economy is facing an increasing global competition and can only remain competitive by strengthening its scientific and technological base. Therefore, one of the key aims of EU policies over recent decades has been to encourage a greater investment in R&D (Ahn 2002).

Despite the EUs long-standing 3% target, the EUs R&D intensity has grown only modestly over the past 20 years. After a prolonged stagnation between 2000 and 2007, the EUs R&D intensity has increased slowly, stabilizing at just above 2.0% since 2011, and reaching 2.3% in 2020. In absolute terms, this corresponded to the R&D expenditure of about EUR 311 billion in 2020, compared with EUR 228 billion in 2011. With the gap of 0.7 percentage points, the EU nevertheless remains at some distance from its ambition of raising the R&D intensity to 3% by 2030 (monitoring report on the progress towards the SDGs in an EU context, Sustainable Development in the European Union 2022).

The labour productivity in the Ukrainian economy is influenced by the factors of the innovative activity group (Figure 4):

$X_{18\_r\&d}$: expenditure on scientific research and development decreased by 0.4% from 2014 (EUR 493.3 million) to 2020 (EUR 491.4 million);

$X_{18\_eie}$: expenditure on innovations of industrial enterprises increased by 3.9% from 2014 (EUR 400.1 million) to 2020 (EUR 415.9 million);

$X_{18\_mes}$: the purchase of machinery, equipment, and software increased by 29.1% from 2014 (EUR 266.0 million) to 2020 (EUR 343.5 million) and had maximum values in 2015 (EUR 424.9 million), 2016 (EUR 697.6 million), and 2019 (EUR 394.6 million).

$X_{18\_r\&die}$: expenses of industrial enterprises on research and development increased by 10.4% from 2014 (EUR 91.2 million) to 2020 (EUR 100.7 million).

However, the costs of getting new technologies and acquiring other external knowledge ($X_{18\_nt}$) in the Ukrainian economy decreased by 64.0% from EUR 2.5 million (2014) to EUR 0.9 million (2020). This is explained by the fact that industrial enterprises spend more money on research and development than on the acquisition of new technologies.

The number of employees involved in scientific research and development in Ukraine decreased by 42.1%.

The study of the factor $X_{17}$, which is the number of new technological processes (units) implemented in production, has a tendency to increase over the last 7 years by 31.3%.

The maximum indicators were reached in 2016 (3489 units), 2019 (2318.0 units), and 2020 (2287.9 units).

The factor $X_{18}$, which is the number of the introduced innovative types of products, names (units) increased by 11.1%, from 3661 units (2014) to 4066 units (2020).

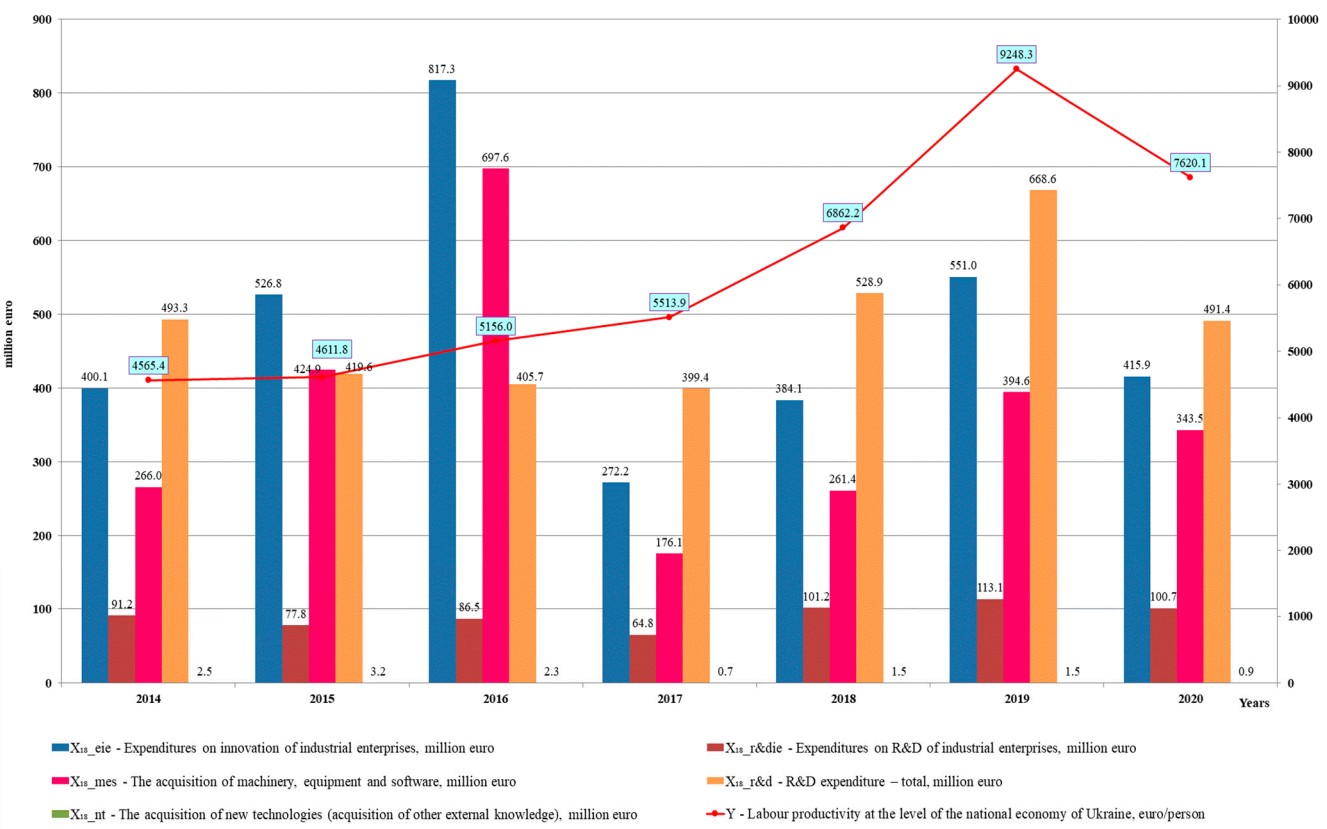

**Figure 4.** Innovation activity in the Ukrainian economy, in 2014–2020. Source: calculated and plotted by the authors according to data (European Commission Eurostat 2022); (Organisation for Economic Co-Operation and Development 2022); (World Bank 2022); (State Statistics Service of Ukraine 2022).

When studying the impact of the group of factors the "Formation and use of personnel" on the growth of labour productivity in EU-27 countries, in 2014–2021, the most influential factors were the following (Appendix A) (Drucker 2003); (Libanova 2019); (Organisation for Economic Co-Operation and Development 2022); (State Statistics Service of Ukraine 2022).

- the specific weight of the employed population that has a full higher education (master's degree) ($X_{31}$) increased by 2.0 p.p. during the specified period and reached maximum values in 2017 and 2018 of 87.9%, 88.2% in 2019, and 88.7% in 2021;
- The share of the employed population with a basic higher education (bachelor's degree) ($X_{30}$) increased by 2.6 p.p., from 81.9% in 2014 to 84.5% in 2021;
- The level of participation in the education and training of the employed population ($X_{31\_pret}$) at the level of the economies of EU-27 countries increased by 0.8 p.p., from 11.6% (2014) to 12.4% (2021);
- The share of enterprises that provide training to their employees ($X_{31\_sept}$) also had an upward trend and increased by 3.6 p.p., from 81.9% (2014) to 85.5% (2021).

Enterprises in EU-27 countries are engaged in improving qualifications, training their employees in new professions, sending their employees to educational institutions, and organizing continuous education. This can significantly improve the performance of the enterprise, since it is qualified employees who have a higher productivity (monitoring report on progress towards the SDGs in an EU context, Sustainable Development in the European Union 2022).



The professional development of specialists, which is a set of organizational and economic measures is related to: training, retraining, and advancing the training of personnel; the organization of inventive and rationalizing work; professional adaptation; the assessment of candidates for a vacant position; the current periodic assessment of personnel; business career planning; working with reserve personnel, etc. At the same time, a company's development strategy and the level of professionalism of each employee are taken into account.

Education makes an employee capable of work; he receives a higher pay for his work results which increases the labour productivity in a specific workplace. Thus, as the education of the workforce increases, the average level of labour productivity in the economy becomes higher (Casu et al. 2004).

The social productivity of labour increases due to the improvement of the quality of education and accordingly, economic growth is stimulated (Antikainen and Lönnqvist 2006).

The level of education determines both the speed of the spread of discoveries and the speed of their direct implementation. This is due to the fact that, first, a further development of science and technology is unthinkable without highly qualified researchers and engineers, scientists who are generators of ideas, and on whom the practical implementation of the discoveries depends. Second, a significant amount of scientific development is carried out within educational institutions. Third, many technological and organizational innovations in production are carried out directly in the workplace. If education makes an employee inventive and proactive and it develops innovative abilities in him, then this contributes to scientific and technical progress, which, in turn, increases social labour productivity (Semiv 2013).

The achievements of scientific and technical progress are able to be understood and practically implemented by employees with a higher education. For example, they are active in technical creativity 30–80 times more than employees with a primary education.

Education remains the most important factor influencing labour productivity and a stimulator of human development.

When studying the impact of the group of factors "Formation and use of personnel" on the growth of labour productivity in the Ukrainian economy for the period 2014–2021, the most influential ones were the following (Figure 5):

- The specific weight of the employed population that has a full higher education (master's degree or Specialist) ($X_{31}$) has increased over the last 8 years by 4 percentage points. The maximum values of this indicator were recorded in 2015 to be 71.9%, in 2019 to be 72.5%, and in 2021 to be 72.7%;
- The share of the employed population with a basic higher education (bachelor's degree) ($X_{30}$) reached maximum values in 2015 and 2018 of 49.3%, 56.9% in 2019, and 57.2% in 2021, which also ensured an increase in labour productivity.

There was a decrease in the specific weight of the employed population with an incomplete higher education (junior specialist or junior bachelor's degree), ($X_{32}$) by 3.6 percentage points from 63.1% (2014) to 59.5% (2021).

When studying the influence of the group of factors the "State of use of fixed capital" on increasing the labour productivity in EU-27 countries in 2014–2021, the following trends are observed:

- The fixed assets in the economy ($X_1$) increased by 17.2% from EUR 2,716,619.6 million (2014) to EUR 318,4247.9 million (2021);
- The specific weight of the active part of the fixed capital ($X_{\_sap}$) decreased by 0.5 percentage points, from 79.021% (2014) to 78.494% (2021);
- The specific weight of machinery and equipment in the active part of the fixed capital ($X_2$) decreased by 0.8 percentage points, from 60.552% (2014) to 59.765% (2021);
- The capital–labour ratio ($X_3$) increased by 27.6% from EUR 11,954.301/employee (2014) to EUR 15,248.236/employee (2021);
- The technological equipment of labour ($X_4$) increased by 26.7% from 9446.5 EUR/employee (2014) to 11,970.540 EUR/employee (2021);

- The machine equipment of labour ($X_5$) increased by 25.1% from 5719.983 EUR/employee (2014) to 7154.164 EUR/employee (2021);
- The coefficient of intellectualization of the fixed capital ($X_8$) increased by 0.8 percentage points, from 19.655% (2014) to 20.489% (2021) (Appendix A, Table A2).

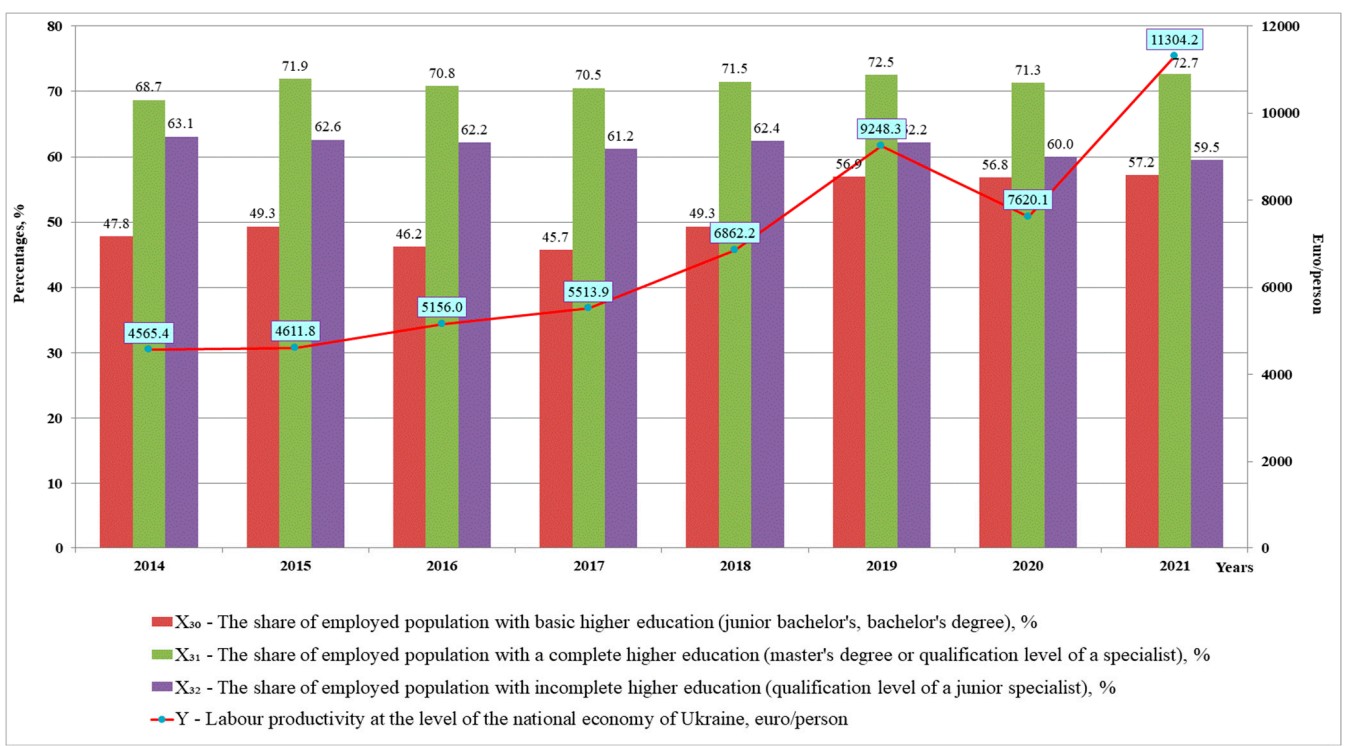

**Figure 5.** Formation and use of personnel in the Ukrainian economy, in 2014–2021. Source: calculated and plotted by the authors according to data (European Commission Eurostat 2022); (Organisation for Economic Co-Operation and Development 2022); (World Bank 2022); (State Statistics Service of Ukraine 2022).

The experience of European Union countries shows that labour productivity increases with the growth of the capital–labour ratio, the technological equipment of labour, and the machine equipment of labour. The technical re-equipment of production, the introduction of the latest technologies, equipment, production automation, and the renewal and replacement of outdated equipment leads to an increase in the cost of fixed capital and jobs. Modern high-tech equipment allows for a more efficient use of labour resources, replacing live labour with machine labour, which leads to an increase in labour productivity (Khirivskyi et al. 2022).

A positive phenomenon is the increase in the intellectualization coefficient of the fixed capital in EU-27 countries, which shows the growth of the share of intangible assets in fixed capital, namely software and databases; the rights to commercial designations and the objects of industrial property; and the copyright and related rights, patents, licenses, concessions, etc. The dynamics of the growth rate of the intellectualization of fixed capital at the level of the economies of EU-27 countries in 2014–2021 has a positive effect on labour productivity.

When studying the influence of the group of factors the "State of use of fixed capital" on labour productivity in the Ukrainian economy for the last 7 years (2014–2020), the following patterns (Appendix A, Table A3) are observed in relation to the factors:

$X_1$: fixed assets in the economy from 2014 to 2020 decreased by EUR 399336,1 million, or by 55,8%. This, in turn, led to a decrease in the indicator $X_3$, the capital–labour ration in the economy, by 50.0% from EUR 39.6 thousand (2014) to EUR 19.8 thousand (2020). In

turn, the removal of physically and morally outdated fixed assets from the total value of fixed assets affected the reduction in the indicator of labour capital;

$X_7$: the coefficient of wear decreased by 25.0 percentage points from 2014 to 2020 (from 83.5% to 58.5%). This is a positive phenomenon, since the untimely replacement of morally obsolete basic production means leads to the fact that the cost price increases and the quality decreases, compared to products made on more advanced machines and equipment;

$X_6$: the renewal coefficient of fixed assets reflects the share of introduced new fixed assets in the total value of fixed assets. Thus, the rate of renewal of fixed assets in the Ukrainian economy increased by 5.4 percentage points, from 2.2% (2014) to 7.6% (2020). Fundamentally new equipment and materials, advanced technologies, machines and units with a higher unit capacity and productivity, complete systems of complex mechanization, and the automation of production are being introduced into production.

The next group of factors, the "Investment activity", at the level of the economies of EU-27 countries is shown in Appendix A, Table A2.

The increase in labour productivity in EU-27 countries is influenced by the indicator of capital investment (investment in tangible and intangible assets), $X_9$, which showed an upward trend from 2014 (EUR 2,716,619.6 million) to 2021 (EUR 3,691,261.827 million) by 35.9%.

Accordingly, the share of investments in GDP also had a tendency to grow by 2.8 percentage points, so in 2014 it was 19.67%, and in 2021 it was 22.43%. Capital investments consist of investments in fixed capital; investments in the capital construction, expansion, and reconstruction of existing buildings and structures; investments in machines, equipment, and inventory; and investments in intangible fixed capital assets.

Businesses were the biggest investors in 2020, with an investment share in GDP of 13.7%, followed by households with 5.4% and governments with 3.3%. The investment share of households has been growing slowly since 2016 but still remains below the levels seen before the 2008 financial crisis. Government investment has followed a counter-cyclical pattern, increasing during both the financial crisis of 2008 and the COVID-19 crisis in 2020.

The influence of the following group of factors on labour productivity at the level of the national economy of Ukraine (investment activity) is shown in Appendix A, Table A3.

$X_9$: capital investments (investments in tangible and intangible assets) increased by 50.6% from 2014 (EUR 11408.57 million) to 2021 (EUR 17182.3 million).

Capital investments are funds directed to the reproduction of fixed assets, the expansion, reconstruction, and modernization of enterprises and structures, the implementation of technical progress in all branches of the economy, the construction of housing, schools, hospitals, and other objects of a social and cultural purpose, and geological exploration and design works.

However, despite such a growth, $X_{10}$, the specific weight of investments in fixed capital to the total volume in the national economy over the last 8 years showed a tendency to decrease by 14.5 percentage points (from 92.7% in 2014 to 78.2% in 2021).

The reduction in the specific weight of the investments in machinery, equipment, and inventory to the total volume ($X_{12}$) at the national level by 1.4 percentage points has a negative effect on the reserves of labour productivity growth from 31.4% (2014) to 30.0% (2021). The final results of the enterprise depend on investment in the material and technical base of production. It is impossible to achieve a high labour productivity on equipment which is outdated and constantly in need of capital repairs.

The specific weight of the investments in capital construction to the total volume at the national level ($X_{11}$) in 2014–2021 showed a tendency to decrease by 17.1 percentage points (from 55.0% to 37.9%). Investments should be directed to new capital construction, and the expansion and reconstruction of existing buildings and structures, which are carried out by contracting and economic methods.

The specific weight of the capital repair investments to the total volume ($X_{14}$) at the level of the national economy in 2014 was 7.0%, and 11.3% in 2021, meaning it increased

by 4.3 percentage points. The large costs for capital repairs are associated with the accumulation of outdated equipment at enterprises. The analysis shows that the use of outdated equipment not only increases the cost of production and reduces competitiveness on international markets, but also narrows the possibilities of introducing new equipment and artificially slows down the pace of scientific and technological progress.

The coefficient of the intellectualization of investments in fixed capital ($X_{15}$) shows the ratio of investments directed to intangible assets as investments in fixed capital, that is, what the specific weight of costs for software and databases are; the rights to commercial designations and objects of industrial property; and copyright and the related rights, patents, licenses, concessions, etc. The dynamics of the intellectualization coefficient of investments in fixed capital in the Ukrainian economy over the past 8 years has a positive trend. For example, in 2014 it was 3.5%, and already in 2021 it increased to 5.7%, meaning it increased by 2.2 percentage points. The maximum value of this indicator was 7.3% (in 2020). One of the powerful reserves for increasing labour productivity is the direction of investments in intangible assets of fixed capital (Roth 2019).

The following group of factors, the "Use of working time", in the economies of EU-27 countries for 2014–2021 should be considered using the following indicators (Appendix A, Table A2):

- The coefficient of part-time employment due to training and education ($X_{22ptet}$) increased by 2.1 p.p., from 10.3% (2014) to 12.4% (2021). As a result, more qualified and qualitatively trained personnel are employed to manufacture products and provide services, which has a positive effect on labour productivity;
- The rate of part-time employment due to the lack of the main job ($X_{22ptelpe}$) decreased by 6.3 p.p. from 29.6% (2014) to 23.3% (2021);
- The ratio of part-time employment in relation to total employment ($X_{22pter}$) decreased by 1.9 percentage points, from 19.6% (2014) to 17.7% (2021).

In EU-27 countries, productive employment is observed as economically beneficial and expedient, which corresponds to the labour potential and the qualifications and abilities of employees and allows them to realize their potential and receive high earnings.

All this indicates the positive growth dynamics of the labour productivity indicator in the economies of EU-27 countries, the maximum values of which were in 2019 68,179.1 EUR/person, and in 2021, 69186.1 EUR/person.

The study of the "Composition of the payroll budget" group in the Ukrainian economy for the period 2014–2021 (Appendix A, Table A3) shows the positive impact of the following factors on increasing labour productivity:

- The specific weight of the additional salary in the payroll budget ($X_{26}$) increased by 3.1 percentage points, from 34.2% (2014) to 37.3% (2021), namely the additional payments, allowances, guarantees, and compensation payments provided by the current legislation and the bonuses related to the performance of production tasks and functions;
- The specific weight of the incentive and compensation payments in the payroll budget ($X_{27}$) increased by 1.3 percentage points, from 5.0% (2014) to 6.3% (2021). These are rewards and bonuses that have a one-time nature, compensatory and other monetary and material payments that are not provided by acts of current legislation, or that are carried out in excess of those established by the specified normative acts;
- The average monthly nominal salary of one employee ($X_{29}$) increased by 151.7%, from EUR 180.9 (2014) to EUR 455.4 (2021). The rate of growth of labour productivity over the rate of growth of the average monthly salary has a fluctuating nature, that leads to a disproportion between the social and economic indicators, slows down the development of production and the production of competitive products.

Factors that do not affect the growth of labour productivity in the Ukrainian economy were also identified:

- The specific weight of basic salary in the payroll budget decreased by 4.4 percentage points, from 60.8% (2014) to 56.4% (2021). The basic salary includes the remuneration for work performed in accordance with the established labour standards (the standards of time, production, service, and job duties);
- The specific weight of payment for unworked time in the payroll budget decreased by 0.6 percentage points, from 9.3% (2014) to 8.7% (2021).

## 4. Conclusions and Empirical Implications

The article offers a practical solution to the scientific objective, which consists of deepening the methodological provisions and developing scientific and practical recommendations for increasing labour productivity in the economies of EU-27 counties and the Ukrainian economy. This made it possible to form the following conclusions:

1. In order to provide a comprehensive assessment of labour productivity in the EU-27 economies and the Ukrainian economy, the system of influencing factors has been developed. They are combined into the following groups: innovative activity; the formation and use of personnel; the state of use of fixed capital; the composition of the payroll budget; the investment activity; and the use of working time. We consider these factors to be decisive in the context of ensuring the possibility of creating added value and innovative development.

2. Based on the use of the method of linearization of the model, rating of the influence of factors on labour productivity in the EU-27 economies and Ukrainian economy was determined. The available reserves for its improvement were clarified and recommendations were given for their development.

3. It was established that a significant positive impact on labour productivity in the EU-27 economies and Ukrainian economy is carried out by several groups of factors: "Innovative activity", the "Formation and use of personnel", and the "State of use of fixed capital". The groups of factors that have insignificant positive impact include: the "Composition of the payroll budget", "Investment activity", and the "Usage of working time".

4. The potential reserves of labour productivity growth for both the economies of the European Union-27 and Ukraine are:

   a. In the "Innovative activity" group: an increase in costs for carrying out scientific research and development; the growth of enterprises R&D expenditures in high-tech sectors; increasing the share of R&D personnel and researchers in the total number of the employed population; the growth of costs for innovation of industrial enterprises; the introduction to the production of new technological processes and innovative types of products; and the purchase of machines, equipment, and software.

   b. In the group the "Formation and use of personnel": an increase in the specific weight of the employed population that has a basic higher education (bachelor's degree); a growth in the share of the employed population that has a full higher education (master's degree); and increasing the share of enterprises that provide training.

   c. In the group the "State of use of fixed capital": an increase in the capital–labour ratio; the technological equipment of labour; the machine equipment of labour; the renewal of fixed capital; and increasing the level of intellectualization of fixed capital.

   d. In the group the "Composition of the payroll budget": an increase in the specific weight of the additional salary in the wage fund; incentive and compensation payments in the payroll budget; and the average monthly nominal salary of one employee.

   e. In the "Investment activity" group: an increase in the capital investment and the share of investment in GDP; increasing the specific weight of investments in

machinery, equipment, and the inventory; and the growth of intellectualization of investments in fixed capital.

It is proven that the management of labour productivity should be carried out through its strategic planning, monitoring, and control. The primary task of the governments of European countries should be the development of a comprehensive program for increasing labour productivity as a factor of ensuring innovation and human development, which should combine the already existing programs for the provision of the individual components and be subordinate to the general program.

**Author Contributions:** Conceptualization, T.K.; methodology, T.K.; software, T.K. and N.T.; validation, T.K., A.L.L., N.T. and M.V.; formal analysis, T.K., A.L.L., N.T. and M.V; investigation, T.K., A.L.L. and N.T.; resources, T.K., A.L.L., N.T. and M.V; data curation, T.K.; writing—original draft preparation, T.K., A.L.L. and N.T.; writing—review and editing, T.K., A.L.L., N.T. and M.V; visualization, T.K., A.L.L., N.T. and M.V; supervision, T.K.; project administration, T.K.; funding acquisition: not applicable. All authors have read and agreed to the published version of the manuscript.

**Funding:** This research received no external funding.

**Institutional Review Board Statement:** Not applicable.

**Informed Consent Statement:** Not applicable.

**Data Availability Statement:** Data are available when required to correspondent author.

**Conflicts of Interest:** The authors declare no conflict of interest.

## Appendix A. Preliminary Tests Tables

**Table A1.** Indicators of labour productivity in the European Union-27 countries and in the Ukrainian economy, 2014–2021.

| Indicator | Unit of Measure | Years | | | | | | | | Deviation, 2021/2014 | |
|---|---|---|---|---|---|---|---|---|---|---|---|
| | | 2014 | 2015 | 2016 | 2017 | 2018 | 2019 | 2020 | 2021 | Absolute, +/− | Relative, % |
| 1 | 2 | 3 | 4 | 5 | 6 | 7 | 8 | 9 | 10 | 11 | 12 |
| Labour productivity at the level of the economies of the European Union—27 countries | EUR/person | 62,014.1 | 64,700.6 | 64,411.4 | 65,316.6 | 66,561.8 | 68,179.1 | 64,909.0 | 69,186.1 | 7172.0 | 11.6 |
| Labour productivity at the level of the national economy of Ukraine | EUR/person | 4565.4 | 4611.8 | 5156.0 | 5513.9 | 6862.2 | 9248.3 | 7620.1 | 11,304.2 | 6738.8 | 147.6 |

Source: own study based on the European Commission Eurostat (2022), State Statistics Service of Ukraine (2022).

**Table A2.** Factors affecting labour productivity in the European Union-27 countries, 2014–2021.

| Factor | Factor by Groups | Unit of Measure | Years | | | | | | | | Deviation, 2021/2014 | |
|---|---|---|---|---|---|---|---|---|---|---|---|---|
| | | | 2014 | 2015 | 2016 | 2017 | 2018 | 2019 | 2020 | 2021 | Absolute, +/− | Relative % |
| 1 | 2 | 3 | 4 | 5 | 6 | 7 | 8 | 9 | 10 | 11 | 12 | 13 |
| **State of use of fixed capital (FC)** | | | | | | | | | | | | |
| $X_1$ | Fixed assets in the economy at actual prices | million EUR | 2,716,619.6 | 2,924,987.8 | 2,996,302.8 | 3,139,934.0 | 3,291,486.3 | 3,530,599.2 | 2,947,352.8 | 3,184,247.9 | 467,628.3 | 17.2 |
| $X_{1sap}$ | The share of the active part of fixed capital | % | 79.0 | 80.5 | 81.6 | 81.6 | 80.5 | 79.5 | 79.3 | 78.5 | −0.5 | - |
| $X_2$ | The share of machinery and equipment in the active part of fixed capital | % | 60.6 | 58.9 | 58.8 | 58.9 | 59.4 | 57.6 | 57.3 | 59.8 | −0.8 | - |
| $X_3$ | Capital–labour ratio | EUR/employed | 11,954.3 | 12,738.6 | 12,879.2 | 13,292.0 | 13,744.9 | 14,595.9 | 14,277.2 | 15,248.2 | 3293.9 | 27.6 |
| $X_4$ | Technological equipment of labour | EUR/employed | 9446.5 | 10,255.1 | 10,508.8 | 10,849.0 | 11,060.5 | 11,601.6 | 11,319.5 | 11,970.5 | 2524.1 | 26.7 |
| $X_5$ | Machine equipment of labour | EUR/employed | 5719.9 | 6047.6 | 6175.2 | 6388.1 | 6570.6 | 6687.4 | 6482.6 | 7154.2 | 1434.2 | 25.1 |
| $X_8$ | Intellectualization coefficient of fixed capital | % | 19.7 | 21.0 | 21.1 | 20.8 | 20.3 | 21.6 | 22.9 | 20.5 | 0.8 | - |
| **Investment activity** | | | | | | | | | | | | |
| $X_9$ | Capital investment, total | million EUR | 2,716,619.6 | 2,924,987.8 | 2,996,302.8 | 3,139,934.0 | 3,291,486.3 | 3,530,599.2 | 3,522,196.4 | 3,691,261.8 | 974,642.2 | 35.9 |
| $X_{9si}$ | Share of investment in GDP | % | 19.7 | 20.1 | 20.4 | 20.8 | 21.1 | 21.9 | 22.3 | 22.4 | 2.8 | - |
| **Innovation activity** | | | | | | | | | | | | |
| $X_{18r\&d}$ | Research and development (internal R&D) | million EUR | 286,510.5 | 303,199.5 | 306,130.9 | 321,194.9 | 336,822.6 | 352,209.8 | 351,364.5 | - | 64,854.0 | 22.6 |
| $X_{18sr\&d}$ | Share of R&D expenditures in GDP, % | % from GDP | 2.000 | 2.010 | 1.990 | 2.030 | 2.070 | 2.100 | 2.190 | - | 0.19 | - |
| $X_{18ber\&d}$ | Business enterprises R&D expenditure in high-tech sectors | million EUR | 182,357.4 | 194,268.3 | 199,433.9 | 211,117.1 | 224,408.2 | 235,628.6 | 232,094.1 | - | 49,736.8 | 27.3 |
| $X_{18ems}$ | Enterprises in high-tech manufacturing sectors | units | 2,120,000.0 | 2,096,668.0 | 212,0592.0 | 2,143,919.0 | 2,171,790.0 | 2,197,851.0 | 2,184,820.5 | - | 64,820.5 | 3.1 |
| $X_{18\_r\&dehtse}$ | R&D expenditures in high-tech sectors per enterprise | million EUR | 86,000.0 | 93,000.0 | 94,000.0 | 98,000.0 | 103,000.0 | 107,000.0 | 106,230.0 | - | 20,230 | 23.5 |
| $X_{18\_sr\&dps}$ | Share of R&D personnel and researchers in total employed population | % | 1.319 | 1.346 | 1.367 | 1.419 | 1.476 | 1.509 | 1.559 | - | 0.24 | - |
| $X_{18\_r\&dpr}$ | R&D personnel and researchers who are fully employed in the economy. individuals | employed | 2,399,423.0 | 2,477,699.0 | 2,556,167.0 | 2,692,860.0 | 2,831,817.0 | 2,921,544.0 | 2,964,580.0 | - | 565,157.0 | 23.6 |
| **Use of working time** | | | | | | | | | | | | |
| $X_{22\_ptet}$ | The coefficient of part-time employment due to education, training | % | 10.3 | 10.4 | 10.7 | 11.3 | 11.6 | 11.9 | 11.3 | 12.4 | 2.1 | - |
| $X_{22\_ptelpe}$ | The coefficient of part-time employment due to the lack of a principal place of employment | % | 29.6 | 29.1 | 27.7 | 26.4 | 24.8 | 23.6 | 24.5 | 23.3 | −6.3 | - |
| $X_{22\_pter}$ | The coefficient of part-time employment in relation to the total employment | % | 19.6 | 19.6 | 19.5 | 19.4 | 19.2 | 19.1 | 18.2 | 17.7 | −1.9 | - |
| $X_{22\_ewtse}$ | The coefficient employment of working time of skilled employees to all employees in the industry | % | 16.4 | 16.3 | 16.2 | 16.3 | 16.3 | 16.1 | 16.2 | 16.3 | −0.1 | - |
| **Formation and use of personnel** | | | | | | | | | | | | |
| $X_{30}$ | Share of employed population with basic higher education (bachelor's degree) | % | 81.9 | 82.6 | 82.9 | 83.7 | 84.1 | 84.3 | 83.8 | 84.5 | 2.6 | - |
| $X_{31}$ | Share of employed population with a complete higher education (master's degree) | % | 86.7 | 87.0 | 87.3 | 87.9 | 87.9 | 88.2 | 87.6 | 88.7 | 2.0 | - |
| $X_{31\_pret}$ | Participation rate in education and training by employed population | % | 11.6 | 11.4 | 11.4 | 11.5 | 11.8 | 12.0 | 12.3 | 12.4 | 0.8 | - |
| $X_{31\_ept}$ | Share of enterprises providing training | % | 81.9 | 82.7 | 84.6 | 84.0 | 84.2 | 85.0 | 85.3 | 85.5 | 3.6 | - |

Source: own study based on the European Commission Eurostat (2022), Organisation for Economic Co-Operation and Development (2022), World Bank (2022).

**Table A3.** Factors affecting labour productivity in the Ukrainian economy, 2014–2021.

| Factor | Factor by Groups | Unit of Measure | 2014 | 2015 | 2016 | 2017 | 2018 | 2019 | 2020 | 2021 | Deviation 2021/2014 Absolute, +/− | Relative, % |
|---|---|---|---|---|---|---|---|---|---|---|---|---|
| 1 | 2 | 3 | 4 | 5 | 6 | 7 | 8 | 9 | 10 | 11 | 12 | 13 |
| | **State of use of fixed capital (FC)** | | | | | | | | | | | |
| $X_1$ | Fixed assets in the economy at actual prices | million EUR | 715,030.9 | 291,397.9 | 287,708.0 | 230,894.5 | 303,019.8 | 382,097.4 | 315,694.8 | - | −399,336.1 | −55.8 |
| $X_3$ | Capital–labour ratio | thousand EUR/employed | 39.6 | 17.7 | 17.7 | 14.3 | 18.5 | 23.0 | 19.8 | - | −19.8 | −50.0 |
| $X_6$ | Renewal coefficient | % | 2.2 | 14.5 | 6.1 | 6.0 | 6.1 | 8.5 | 7.6 | - | 5.4 | - |
| $X_7$ | Coefficient of wear | % | 83.5 | 60.1 | 58.1 | 55.1 | 60.6 | 56.4 | 58.5 | - | −25.0 | - |
| | **Investment activity** | | | | | | | | | | | |
| $X_9$ | Capital investment, total | million EUR | 11,408.57 | 10,415.1 | 12,638.4 | 12,324.5 | 16,596.5 | 22,645.6 | 14,672.5 | 17,182.3 | 5773.7 | 50.6 |
| $X_{10}$ | The share of investments in fixed capital to the total amount | % | 92.7 | 89.3 | 92.3 | 92.3 | 89.4 | 91.6 | 66.8 | 78.2 | −14.5 | - |
| $X_{11}$ | The share of investments in capital construction to the total amount | % | 55.0 | 51.2 | 47.8 | 44.8 | 44.2 | 46.7 | 25.2 | 37.9 | −17.1 | - |
| $X_{12}$ | The share of investments in machinery. equipment and inventory to the total amount | % | 31.4 | 30.9 | 34.3 | 34.8 | 33.1 | 34.0 | 31.8 | 30.0 | −1.4 | - |
| $X_{14}$ | The share of investments in capital repairs to the total amount | % | 7.0 | 7.7 | 7.9 | 9.4 | 9.7 | 10.8 | 11.7 | 11.3 | 4.3 | - |
| $X_{15}$ | The coefficient of intellectualization of fixed capital investment | % | 3.5 | 7.0 | 3.4 | 3.9 | 7.0 | 3.8 | 7.3 | 5.7 | 2.2 | - |
| | **Innovation activity** | | | | | | | | | | | |
| $X_{17}$ | The number of new technological processes introduced into production | units | 1743 | 1217 | 3489 | 1831 | 2002 | 2318.0 | 2287.9 | - | 544.9 | 31.3 |
| $X_{18}$ | The number of introduced innovative types of products, by names | units | 3661 | 3136 | 4139 | 2387 | 3843 | 2148 | 4066 | - | 405 | 11.1 |
| $X_{18r\&d}$ | R&D expenditure—total | million EUR | 493.3 | 419.6 | 405.7 | 399.4 | 528.9 | 668.6 | 491.4 | - | −1.9 | −0.4 |
| $X_{18sr\&d}$ | Share of R&D expenditures in GDP. % | % | 0.60 | 0.55 | 0.48 | 0.45 | 0.47 | 0.43 | 0.41 | - | −0.19 | - |
| $X_{18eie}$ | Expenditures on innovation of industrial enterprises | million EUR | 400.1 | 526.8 | 817.3 | 272.2 | 384.1 | 551.0 | 415.9 | - | 15.8 | 3.9 |
| $X_{18r\&die}$ | Expenditures on R&D of industrial enterprises | million EUR | 91.2 | 77.8 | 86.5 | 64.8 | 101.2 | 113.1 | 100.7 | - | 9.5 | 10.4 |
| $X_{18nt}$ | The acquisition of new technologies (acquisition of other external knowledge) | million EUR | 2.5 | 3.2 | 2.3 | 0.7 | 1.5 | 1.5 | 0.9 | - | −1.6 | −64.0 |
| $X_{18mes}$ | The acquisition of machinery, equipment and software | million EUR | 266.0 | 424.9 | 697.6 | 176.1 | 261.4 | 394.6 | 343.5 | - | 77.5 | 29.1 |
| $X_{18r\&dp}$ | Number of R&D personnel—total | persons | 136,123 | 122,504 | 97,912 | 94,274 | 88,128 | 79,262 | 78,860 | - | −57,263 | −42.1 |
| | **Composition of the payroll budget** | | | | | | | | | | | |
| $X_{25}$ | The share of basic salary in the payroll budget | % | 60.8 | 57.8 | 58.8 | 58.4 | 57.5 | 57.1 | 56.6 | 56.4 | −4.4 | - |
| $X_{26}$ | The share of additional salary in the payroll budget | % | 34.2 | 37.1 | 35.8 | 36.1 | 36.9 | 37.2 | 37.7 | 37.3 | 3.1 | - |
| $X_{27}$ | The share of incentive and compensation payments in the payroll budget | % | 5.0 | 5.0 | 5.5 | 5.7 | 5.5 | 5.6 | 5.7 | 6.3 | 1.3 | - |
| $X_{28}$ | The share of payment for the time not worked in the payroll budget | % | 9.3 | 8.6 | 9.2 | 7.9 | 8.3 | 8.5 | 8.9 | 8.7 | −0.6 | - |
| $X_{29}$ | The average monthly nominal salary per employee | EUR | 180.9 | 159.9 | 182.4 | 212.1 | 279.5 | 406.7 | 334.6 | 455.4 | 274.5 | 151.7 |
| | **Formation and use of personnel** | | | | | | | | | | | |
| $X_{30}$ | The share of employed population with basic higher education (junior bachelor's, bachelor's degree) | % | 47.8 | 49.3 | 46.2 | 45.7 | 49.3 | 56.9 | 56.8 | 57.2 | 9.4 | - |
| $X_{31}$ | The share of employed population with a complete higher education (master's degree or qualification level of a specialist) | % | 68.7 | 71.9 | 70.8 | 70.5 | 71.5 | 72.5 | 71.3 | 72.7 | 4.0 | - |
| $X_{32}$ | The share of employed population with incomplete higher education (qualification level of a junior specialist) | % | 63.1 | 62.6 | 62.2 | 61.2 | 62.4 | 62.2 | 60.0 | 59.5 | −3.6 | - |

Source: own study based on the European Commission Eurostat (2022), Organisation for Economic Co-Operation and Development (2022), World Bank (2022), State Statistics Service of Ukraine (2022).

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
