# Peer review of "Assessment of Labour Productivity and the Factors of Its Increase in European Union 27 and Ukrainian Economies"

_economies, doi:10.3390/economies10110287_

Round 1
Reviewer 1 Report
In my opinion the article „Assessment of Labour Productivity and the Reserves of its increase at the level of the European Union-27 Economy and the National Economy of Ukraine” is interesting. However, I would suggest some changes and alterations:
- one cannot make any reserves of productivity. One can talk about productivity in the categories in time in which it can increase or decrease or stay at the same level. If the author introduces a new notion he/she should start with an exact explanation and define it,
- the article lacks any information on what is new in it in comparison to other research,
- the author(s) could think about presenting hypotheses at the beginning of the publication,
- the author(s) should explain why the analysis of the economy of Ukraine has been chosen in comparison to the European Union-27,
- in fig 1 the labour productivity at the level of the national economy of Portugal is presented. It should be explained why the author(s) have chosen Portugal or to remove this country from the picture,
- the example of Greece is presented on page 7. The author(s) should either describe all the 27 countries of the EU or take into consideration the European average,
- fig 2 and fig 2.1 (and 3, 3.1) have the same titles. I suggest differentiating them,
- It is written that “Education remains the most important factor…” (page 11). Probably the author(s) mean that it is “one of the most important factors…”,
- this sentence “The available reserves for its improvement were clarified and recommendations were given for their development” is in fact not justified in the article and requires further explanations,
- I am of the opinion that the references should be enlarged. There are only 20 positions of the references.
Author Response
Dear Reviewer 1, thank you very much for the opportunity to improve our manuscript. We tried to implement all the changes requested, as you can check below. Unfortunately, we encountered unexpected difficulties while working with the Track Changes system. Somehow, we lost the document with the marked corrections. However, we consider that all the revision work done is valid and therefore, in spite of everything, we are sending the paper without the Track Changes system, but with the marked changes. I hope you consider it.
All the best.
The authors.
In my opinion the article „Assessment of Labour Productivity and the Reserves of its increase at the level of the European Union-27 Economy and the National Economy of Ukraine” is interesting. However, I would suggest some changes and alterations:
- one cannot make any reserves of productivity. One can talk about productivity in the categories in time in which it can increase or decrease or stay at the same level. If the author introduces a new notion he/she should start with an exact explanation and define it.
Thank you for your feedback. In our article, we argue that labour productivity can increase or decrease by the influence of different factors, but, at same time, these factors can be reserves for its growth.
Reserves for labour productivity growth are such opportunities for its increase that have already been identified, but for various reasons, they have not yet been used. Reserves are used and re-emerge under the influence of scientific and technological progress. Quantitatively, reserves can be defined as the difference between the maximum possible and actually achieved a level of labour productivity at a specific time. Thus, the use of the reserves of labour productivity growth is the process of transforming the possible into the real.
Since the reserve is actually a segment of the factor that is available for use at a specific moment of time, reserves of labour productivity growth are classified in the same way as factors.
Each type of reserves should be considered according to a single factor, and the totality of reserves should be classified according to the classification of factors. This enables the methodology to identify the main causes of losses and non-productive costs of labour by a different factor and to determine ways to eliminate them.
- the article lacks any information on what is new in it in comparison to other research.
Thank you for your comment. Some new and relevant information has been included in order to overcome this stated limitation. The novelty of the research is the comparative analysis between Ukraine and European Union. Despite numerous studies on productivity in Europe, only now labour productivity in Ukraine is being studied in comparison with benchmark countries. The developed methodology for building a model for calculating the rating coefficients of the influence of development factors on labour productivity can be used at the macro-levels.
- the author(s) could think about presenting hypotheses at the beginning of the publication
Thank you for your comment. Indeed, we could have gone down that route, which is always valid. However, given the approach chosen for this article, it seemed more appropriate not to elaborate on initial hypotheses.
- the author(s) should explain why the analysis of the economy of Ukraine has been chosen in comparison to the European Union-27.
Thank you for the question. Given the particular situation currently experienced in Ukraine, and following the application for membership, in which it achieved the status of a candidate country in June, it is important to understand the factors determining productivity. Post-conflict economic recovery will depend very much on this. Thus, it becomes pertinent to evaluate the indicators associated with the labor productivity of the country compared to the European average.
- in fig 1 the labour productivity at the level of the national economy of Portugal is presented. It should be explained why the author(s) have chosen Portugal or to remove this country from the picture.
Thank you for your comment. We have opted to remove Portugal since that analysis is not essential for the article.
- the example of Greece is presented on page 7. The author(s) should either describe all 27 countries of the EU or take into consideration the European average.
The example of Greece was also removed since it was not relevant to the analysis.
- fig 2 and fig 2.1 (and 3, 3.1) have the same titles. I suggest differentiating them,
Following your suggestion, we correct the figures: 2.1 Innovation activity in high-tech sectors at the level of EU-27 economies in 2014-2020; 3.1. Innovation of industrial enterprises of Ukraine in 2014-2020.
- It is written that “Education remains the most important factor…” (page 11). Probably the author(s) mean that it is “one of the most important factors…”,
Thank you. We´ve corrected the sentence as advised.
- this sentence “The available reserves for its improvement were clarified and recommendations were given for their development” is in fact not justified in the article and requires further explanations.
Thank you for the suggestion. We have tried to justify the text better.
- I am of the opinion that the references should be enlarged. There are only 20 positions of the references.
Thank you for your comment. We have updated the article with other references, such as:
Ahn, Sanghoon (2002). Competition, innovation and productivity growth: a review of theory and evidence. Available at SSRN: http://dx.doi.org/10.2139/ssrn.318059
Astakhova, K. V., Korobeev, A. I., Prokhorova, V. V., Kolupaev, A. A., Vorotnoy, M. V., & Kucheryavaya, E. R. (2016). The role of education in economic and social development of the country. International Review of Management and Marketing, 6(1S).
Bhawsar, P., & Chattopadhyay, U. (2015). Competitiveness: Review, reflections and directions. Global Business Review, 16(4), 665-679.
Buckley, P. J., Pass, C. L., & Prescott, K. (1988). Measures of international competitiveness: a critical survey. Journal of marketing management, 4(2), 175-200.
Blundell, R., Dearden, L., Meghir, C., & Sianesi, B. (1999). Human capital investment: the returns from education and training to the individual, the firm and the economy. Fiscal studies, 20(1), 1-23.
Cherep, A. V., Cherep, O. H., Krylov, D. V., & Voronkovа, V. G. (2019). Methodological approach to the redistribution of investment projects within a company According to formal criteria. Financial and credit activity problems of theory and practice, 1(28), 256-263.
Fagerberg, J. (1996). Technology and competitiveness. Oxford review of economic policy, 12(3), 39-51.
Gibb, A. A. (1993). Enterprise culture and education: understanding enterprise education and its links with small business, entrepreneurship and wider educational goals. International small business journal, 11(3), 11-34.
Gorman, G., Hanlon, D., & King, W. (1997). Some research perspectives on entrepreneurship education, enterprise education and education for small business management: a ten-year literature review. International small business journal, 15(3), 56-77.
Kolot, A. (2020). Labor 4.0 concept: theoretical-applicable principles of formation and development. 1735209643.
Lange, F., & Topel, R. (2006). The social value of education and human capital. Handbook of the Economics of Education, 1, 459-509.
Roth, F. (2019). Intangible capital and labour productivity growth: a review of the literature.
Semykina, M., Luchyk, S., Zapirchenko, L., Semykina, A., Savelenko, H., & Sikoraka, V. (2021, September). Motivativational Mechanism of Activation of Innovative Activity of Personnel and its Improvement. In 2021 11th International Conference on Advanced Computer Information Technologies (ACIT) (pp. 317-321). IEEE.
Shaulska, L., Kovalenko, S., Allayarov, S., Sydorenko, O., & Sukhanova, A. (2021). Strategic enterprise competitiveness management under global challenges. Academy of Strategic Management Journal, 20(4), 1-7.
Tether, B. S., & Hipp, C. (2002). Knowledge intensive, technical and other services: patterns of competitiveness and innovation compared. Technology Analysis & Strategic Management, 14(2), 163-182.
Verdoorn, P.J. (2002). Factors that Determine the Growth of Labour Productivity. In: McCombie, J., Pugno, M., Soro, B. (eds) Productivity Growth and Economic Performance. Palgrave Macmillan, London, 28-36. Retrieved from: https://doi.org/10.1057/9780230504233_2
Yi, W., & Chan, A. P. (2014). Critical review of labor productivity research in construction journals. Journal of Management in Engineering. Volume 30 Issue 2. Retrieved from: https://doi.org/10.1061/(ASCE)ME.1943-5479.0000194
Reviewer 2 Report
First of all, I understand how difficult it must be to work on research in Ukraine under the current conditions, so kudos to the author. Unfortunately, the paper exhibits several flaws which need to be addressed before it can be published:
1. The paper includes big descriptive parts which could be "outsourced" to literature, provided there is a good literature review. If I understood the paper correctly, the idea was to apply the methodology of modeling labor productivity, tested on the data from the EU, on Ukraine, and assess the differences. So, results from labor productivity studies in the EU, especially regarding the used methodology should be presented in the literature review section. Also, it would be a good idea to include some works from the 20th century, not only the 18th and 19th-century ones, and organize them by importance from most influential to least influential.
2. The text should contain citations when something else than the results is described. There is a big chunk on the EU's approach to labor productivity and sustainability without any reference to literature at all.
3. Model and data should be described in sufficient detail: a) it should be explained why this particular model was employed (for example, such approach is prevalent in literature, see point 1 on literature review) and whether it applies to Ukraine (for example, a model tested on Poland and Romania may be perfectly comparable to Ukraine, but Germany-based one may prove to be problematic – if some adjustments were made, one needs to describe them); b) the data should be described as well: sources, completeness (missing values, better a full range of descriptive statistics), periodicity (yearly, quarterly) and how the main variable, labor productivity, was defined (from what I saw, it seems to be GDP per worker or even GDP per capita, which is a questionable choice, but there are alternative definitions like GDP per hour worked); c) what estimator and tests were used to calculate the result – ordinary least squares? if the data are time series, one should think about Engle-Granger tests.
4. Presentation of results is too lengthy and detailed. It would be a good idea to minimize the results section but to prepare a bigger discussion section where the results are compared to previous research from the literature review (currently missing) and data on the EU countries. This would help explain the value added of the paper.
Author Response
Dear Reviewer 2, thank you very much for the opportunity to improve our manuscript. We tried to implement all the changes requested, as you can check below. Unfortunately, we encountered unexpected difficulties while working with the Track Changes system. Somehow, we lost the document with the marked corrections. However, we consider that all the revision work done is valid and therefore, in spite of everything, we are sending the paper without the Track Changes system, but with the marked changes. I hope you consider it.
All the best.
The authors.
First of all, I understand how difficult it must be to work on research in Ukraine under the current conditions, so kudos to the author. Unfortunately, the paper exhibits several flaws which need to be addressed before it can be published:
- The paper includes big descriptive parts which could be "outsourced" to literature, provided there is a good literature review.
Thank you for your comment. We have followed your suggestion and tried to reduce not only the overly descriptive parts but also to improve the literature review.
If I understood the paper correctly, the idea was to apply the methodology of modeling labor productivity, tested on the data from the EU, on Ukraine, and assess the differences. So, results from labor productivity studies in the EU, especially regarding the used methodology should be presented in the literature review section. Also, it would be a good idea to include some works from the 20th century, not only the 18th and 19th-century ones, and organize them by importance from most influential to least influential.
Thank you for your comment, which we appreciate. Several more current labour productivity evaluation methods have been included, knowing that these only determine the impact of certain factors on the efficiency of the use of certain resources. The following references were added that refer to authors who develop methodologies for evaluating labour productivity:
Antikainen, R., & Lönnqvist, A. (2006). Knowledge work productivity assessment. Institute of Industrial Management. Tampere University of Technology. PO Box, 541, 79-102.
Casu, B., Girardone, C., & Molyneux, P. (2004). Productivity change in European banking: A comparison of parametric and non-parametric approaches. Journal of Banking & Finance, 28(10), 2521-2540.
Khirivskyi, R., Yatsiv, I., Petryshyn, L., Pasichnyk, T., Kucher, L., & Irtyshcheva, I. (2022). Assessment of the efficiency of employment of the communities’ resource potential using different approaches. TEM Journal, 11(1), 367-373.
- The text should contain citations when something else than the results is described. There is a big chunk on the EU's approach to labor productivity and sustainability without any reference to literature at all.
Thank you for your comment. We added several citations, regarding studies concerning EU labour productivity such as:
Ahn, Sanghoon (2002). Competition, innovation and productivity growth: a review of theory and evidence. Available at SSRN: http://dx.doi.org/10.2139/ssrn.318059
Astakhova, K. V., Korobeev, A. I., Prokhorova, V. V., Kolupaev, A. A., Vorotnoy, M. V., & Kucheryavaya, E. R. (2016). The role of education in economic and social development of the country. International Review of Management and Marketing, 6(1S).
Bhawsar, P., & Chattopadhyay, U. (2015). Competitiveness: Review, reflections and directions. Global Business Review, 16(4), 665-679.
Buckley, P. J., Pass, C. L., & Prescott, K. (1988). Measures of international competitiveness: a critical survey. Journal of marketing management, 4(2), 175-200.
Blundell, R., Dearden, L., Meghir, C., & Sianesi, B. (1999). Human capital investment: the returns from education and training to the individual, the firm and the economy. Fiscal studies, 20(1), 1-23.
Cherep, A. V., Cherep, O. H., Krylov, D. V., & Voronkovа, V. G. (2019). Methodological approach to the redistribution of investment projects within a company According to formal criteria. Financial and credit activity problems of theory and practice, 1(28), 256-263.
Fagerberg, J. (1996). Technology and competitiveness. Oxford review of economic policy, 12(3), 39-51.
Gibb, A. A. (1993). Enterprise culture and education: understanding enterprise education and its links with small business, entrepreneurship and wider educational goals. International small business journal, 11(3), 11-34.
Gorman, G., Hanlon, D., & King, W. (1997). Some research perspectives on entrepreneurship education, enterprise education and education for small business management: a ten-year literature review. International small business journal, 15(3), 56-77.
Kolot, A. (2020). Labor 4.0 concept: theoretical-applicable principles of formation and development. 1735209643.
Lange, F., & Topel, R. (2006). The social value of education and human capital. Handbook of the Economics of Education, 1, 459-509.
Roth, F. (2019). Intangible capital and labour productivity growth: a review of the literature.
Semykina, M., Luchyk, S., Zapirchenko, L., Semykina, A., Savelenko, H., & Sikoraka, V. (2021, September). Motivativational Mechanism of Activation of Innovative Activity of Personnel and its Improvement. In 2021 11th International Conference on Advanced Computer Information Technologies (ACIT) (pp. 317-321). IEEE.
Shaulska, L., Kovalenko, S., Allayarov, S., Sydorenko, O., & Sukhanova, A. (2021). Strategic enterprise competitiveness management under global challenges. Academy of Strategic Management Journal, 20(4), 1-7.
Tether, B. S., & Hipp, C. (2002). Knowledge intensive, technical and other services: patterns of competitiveness and innovation compared. Technology Analysis & Strategic Management, 14(2), 163-182.
Yi, W., & Chan, A. P. (2014). Critical review of labor productivity research in construction journals. Journal of Management in Engineering. Volume 30 Issue 2. Retrieved from: https://doi.org/10.1061/(ASCE)ME.1943-5479.0000194
- Model and data should be described in sufficient detail: a) it should be explained why this particular model was employed (for example, such approach is prevalent in literature, see point 1 on literature review) and whether it applies to Ukraine (for example, a model tested on Poland and Romania may be perfectly comparable to Ukraine, but Germany-based one may prove to be problematic – if some adjustments were made, one needs to describe them); b) the data should be described as well: sources, completeness (missing values, better a full range of descriptive statistics), periodicity (yearly, quarterly) and how the main variable, labor productivity, was defined (from what I saw, it seems to be GDP per worker or even GDP per capita, which is a questionable choice, but there are alternative definitions like GDP per hour worked); c) what estimator and tests were used to calculate the result – ordinary least squares? if the data are time series, one should think about Engle-Granger tests.
Thank you for your comment. We have followed your suggestions and tried to be more explicit.
The method of building a model for determining the coefficients of rating of the factors of enterprise development using the method of linearization of the model of labour efficiency is proposed.
Based on the use of the method of linearization of labour productivity (y) model to determine the rating of factors (positive: significant, insignificant; negative: significant, insignificant; without impact) calculated average values of the rating coefficients (by the groups)
Currently, the issue of determining the influence of a number of socio-economic, investment, innovation, technical and technological, and organizational factors on the level of labour productivity is very significant.
In order to solve it, it is necessary to develop a methodology for building a model for calculating rating coefficients of the influence of economic development factors at the macro-level on labour productivity.
Modern methods of assessing labour productivity, including taking into account resource potential, were studied. It was found that they determine only the impact of certain factors on the efficiency of using certain resources: correlation relationship, multiple linear regression model, correlation-regression analysis method, Ferrar-Glouber algorithm, Ridge estimation method (i.e. Ridge regression), method of extrapolation, and method of systematization. However, a certain part of theoretical and practical issues related to the determination of the influence of factors of economic development at the macro-level on labour productivity has not yet been fully considered. In particular, the issue of determining the rating of the influence of economic development factors at the macro level on labour productivity has not been sufficiently studied. We introduce the concept of factor rating coefficient, which can be the coefficient for the factor of a linear model. The developed methodology for building a model for calculating the rating coefficients of the influence of development factors on labour productivity can be used at macro, meso, and micro-levels. Labour productivity should be determined at the international level: as the ratio of gross domestic product according to purchasing power parity and the employed population; at the macro level (levels of the national economy): as the ratio of gross domestic product in actual prices and the number of employed population.
- Presentation of results is too lengthy and detailed. It would be a good idea to minimize the results section but to prepare a bigger discussion section where the results are compared to previous research from the literature review (currently missing) and data on the EU countries. This would help explain the value added of the paper.
Thank you for your question. Regarding this comment, some new and relevant information has been included in order to overcome this stated limitation. We have also minimised the results section in order to make it shorter to read and less detailed. At the same time, we have added some references and authors with recent studies in the area to provide a discussion of the results, as suggested.
Reviewer 3 Report
The authors aim to identify the impact of factors on increasing labor productivity at the level of the economies of the European Union-27 and at the level of the national economy 7 of Ukraine. However, there is a doubt if the aim is achieved. The main concern is related with the scientific background of this research. The paper is written mainly in a descriptive style, analyzing the changes of various indicators over 7 years. The authors state that they use the methodology developed by us (Korneeva, 2020) and build a model for calculating the rating coefficients of the influence of factors on labor productivity. Although the model is written, the more detailed methodology about the calculations, made by authors, should be provided. What are the values of rating coefficients of the influence of factors? If a regression analysis is used the issue of multicollinearity should be discussed as lots of related indicators are used in the analysis.
The authors should also think about the novelty of their research as there are lots of research made on labour productivity in EU. However, the literature review is quite poor and just several scientific papers are analyzed in this paper. Although the authors aim to focus on EU and Ukraine, Portugal is also mentioned in the paper and it is not clear why.
Author Response
Dear Reviewer 3, thank you very much for the opportunity to improve our manuscript. We tried to implement all the changes requested, as you can check below. Unfortunately, we encountered unexpected difficulties while working with the Track Changes system. Somehow, we lost the document with the marked corrections. However, we consider that all the revision work done is valid and therefore, in spite of everything, we are sending the paper without the Track Changes system, but with the marked changes. I hope you consider it.
All the best.
The authors.
The authors aim to identify the impact of factors on increasing labor productivity at the level of the economies of the European Union-27 and at the level of the national economy 7 of Ukraine. However, there is a doubt if the aim is achieved. The main concern is related with the scientific background of this research. The paper is written mainly in a descriptive style, analyzing the changes of various indicators over 7 years. The authors state that they use the methodology developed by us (Korneeva, 2020) and build a model for calculating the rating coefficients of the influence of factors on labor productivity. Although the model is written, the more detailed methodology about the calculations, made by authors, should be provided.
Thank you for your comment. We have improved the paper with your suggestion. Many details have been added regarding the elaboration of the model, in order to make the methodological part more explicit.
What are the values of rating coefficients of the influence of factors?
Thank you for the comment. Several tables have been added to clarify the calculations made.
Table 1: l The value of the coefficients of the rating of the influence of economic development factors at the level of the economies of the European Union – 27 countries on labour productivity (y), p.11.
Table 2: The average values of the rating coefficients of the influence of economic development factors at the level of the economies of the European Union-27 countries (by the groups) on labour productivity (y), p.13.
Table 3: The value of the coefficients of the rating of the influence of economic development factors at the level of the national economy of Ukraine on labour productivity. p.14.
Table 4: The average values of the rating coefficients of the influence of economic development factors at the level of the national economy of Ukraine (by the groups) on labour productivity (y). p.16.
If a regression analysis is used the issue of multicollinearity should be discussed as lots of related indicators are used in the analysis.
Thank you for your comment. Multicollinearity can, indeed, be a problem in a regression model because we would not be able to distinguish between the individual effects of the independent variables on the dependent variable. However, in the present paper, the influence of the paired interactions is not taken into account due to the impossibility of managing changes in the factors of economic development in given intervals in real production conditions. Multicollinearity does not affect the accuracy of the model due to the way the values of rating coefficients were determined.
Based on the use of the method of linearization of labour productivity (y) model to determine the rating of factors (positive: significant, insignificant; negative: significant, insignificant; without impact) average values of the rating coefficients (by the groups) were calculated.
The authors should also think about the novelty of their research as there are lots of research made on labour productivity in EU.
Thank you for the comment. The novelty of the research is the comparative analysis between Ukraine and the European Union. Despite numerous studies on productivity in Europe, only now labour productivity in Ukraine is being studied in comparison with benchmark countries.
However, the literature review is quite poor and just several scientific papers are analyzed in this paper. Although the authors aim to focus on EU and Ukraine, Portugal is also mentioned in the paper and it is not clear why.
Thank you for your recommendations.
The example of Portugal was removed since it was not relevant for the analysis.
The literature review is updated by articles like:
Ahn, Sanghoon (2002). Competition, innovation and productivity growth: a review of theory and evidence. Available at SSRN: http://dx.doi.org/10.2139/ssrn.318059
Bhawsar, P., & Chattopadhyay, U. (2015). Competitiveness: Review, reflections and directions. Global Business Review, 16(4), 665-679.
Buckley, P. J., Pass, C. L., & Prescott, K. (1988). Measures of international competitiveness: a critical survey. Journal of marketing management, 4(2), 175-200.
Cherep, A. V., Cherep, O. H., Krylov, D. V., & Voronkovа, V. G. (2019). Methodological approach to the redistribution of investment projects within a company According to formal criteria. Financial and credit activity problems of theory and practice, 1(28), 256-263.
Fagerberg, J. (1996). Technology and competitiveness. Oxford review of economic policy, 12(3), 39-51.
Kolot, A. (2020). Labor 4.0 concept: theoretical-applicable principles of formation and development. 1735209643.
Semykina, M., Luchyk, S., Zapirchenko, L., Semykina, A., Savelenko, H., & Sikoraka, V. (2021, September). Motivativational Mechanism of Activation of Innovative Activity of Personnel and its Improvement. In 2021 11th International Conference on Advanced Computer Information Technologies (ACIT) (pp. 317-321). IEEE.
Shaulska, L., Kovalenko, S., Allayarov, S., Sydorenko, O., & Sukhanova, A. (2021). Strategic enterprise competitiveness management under global challenges. Academy of Strategic Management Journal, 20(4), 1-7.
Tether, B. S., & Hipp, C. (2002). Knowledge intensive, technical and other services: patterns of competitiveness and innovation compared. Technology Analysis & Strategic Management, 14(2), 163-182.
Yi, W., & Chan, A. P. (2014). Critical review of labor productivity research in construction journals. Journal of Management in Engineering. Volume 30 Issue 2. Retrieved from: https://doi.org/10.1061/(ASCE)ME.1943-5479.0000194
Round 2
Reviewer 1 Report
In my opinion, the author should carefully reconsider changing the title of their article "Assessment of Labor Productivity and the Reserves of its increase…". This title, which is presented by the authors, suggests that these reserves come from Labor productivity, which is methodically incorrect. They come from other factors, which have an influence on productivity.
Author Response
Review Report (Round 2)
Dear Reviewer 1, thank you very much for the opportunity to improve our manuscript.
In my opinion, the author should carefully reconsider changing the title of their article "Assessment of Labor Productivity and the Reserves of its increase…". This title, which is presented by the authors, suggests that these reserves come from Labor productivity, which is methodically incorrect. They come from other factors, which have an influence on productivity.
Thank you for your feedback and comment. We have followed your suggestions and reconsidered changing the title of our article. The new proposed title is the following:
“Assessment of labour productivity and the factors of its increase in European Union-27 and Ukrainian economies”. It is more direct and the fundamental idea remains. In line with your suggestion, we have also tried to include all relevant references in the introduction as well as providing sufficient background.
All the best.
The authors.
Reviewer 2 Report
The way the research is presented in the paper has been greatly improved.
One could suggest more improvements but I think the minimum necessary for publication has already been achieved.
I have no further remarks.
Good luck!
Author Response
Review Report (Round 2)
Dear Reviewer 2, thank you very much for the opportunity to improve our manuscript.
The way the research is presented in the paper has been greatly improved.
One could suggest more improvements but I think the minimum necessary for publication has already been achieved.
I have no further remarks.
Good luck!
Thank you very much for your response and for your feedback on improving this article. We have tried to take into account all your suggestions and recommendations. Several changes have been made to the text, in order to improve both form and content, while maintaining the underlying reasoning. The use of (British) English has been improved and made consistent throughout the text.
All the best.
The authors.
Reviewer 3 Report
The content of the paper has been improved, but text formatting (style, spacing, etc.) is needed.
Author Response
Review Report (Round 2)
Dear Reviewer 3, thank you very much for the opportunity to improve our manuscript.
The content of the paper has been improved, but text formatting (style, spacing, etc.) is needed.
Thank you for your recommendations. We have formatted the text as required by the Journal Economies. In this new corrected version, we have tried to considerably improve the formatting, spacing and form of the work, without loss of character or content.
All the best.
The authors.